# Leveraging Artificial Intelligence and Participatory Modeling to Support Paradigm Shifts in Public Health: An Application to Obesity and Evidence-Based Policymaking

**Philippe J. Giabbanelli** [1,*] and **Grace MacEwan** [2]

1   Department of Computer Science & Software Engineering, Miami University, Oxford, OH 45056, USA
2   Department of Family Practice, University of British Columbia, Vancouver, BC V6T 1Z3, Canada
*   Correspondence: giabbapj@miamioh.edu

**Abstract:** The Provincial Health Services Authority (PHSA) of British Columbia suggested that a paradigm shift from weight to well-being could address the unintended consequences of focusing on obesity and improve the outcomes of efforts to address the challenges facing both individuals and our healthcare system. In this paper, we jointly used artificial intelligence (AI) and participatory modeling to examine the possible consequences of this paradigm shift. Specifically, we created a conceptual map with 19 experts to understand how obesity and physical and mental well-being connect to each other and other factors. Three analyses were performed. First, we analyzed the factors that directly connect to obesity and well-being, both in terms of causes and consequences. Second, we created a reduced version of the map and examined the connections between categories of factors (e.g., food production, and physiology). Third, we explored the themes in the interviews when discussing either well-being or obesity. Our results show that obesity was viewed from a medical perspective as a problem, whereas well-being led to broad and diverse solution-oriented themes. In particular, we found that taking a well-being perspective can be more comprehensive without losing the relevance of the physiological aspects that an obesity-centric perspective focuses on.

**Keywords:** AI for decision and policymaking; AI enabling transformation; causal modeling; natural language processing

## 1. Introduction

The majority of Canadians aged 18 and older are either obese (26.8%) or overweight (36.3%), based on the Canadian Community Health Survey [1]. The high level and increasing prevalence (over time) of obesity and overweight is a global phenomenon. Other studies have shown that obesity doubled in 73 countries within a few decades, and the growth was even more pronounced for countries with a low-middle degree of development [2]. The Obesity Society published a position paper, defining obesity as a disease in 2008; its updated position views obesity as a "multi-causal chronic disease" that may be accompanied by 'structural abnormalities' (e.g., musculoskeletal disorders), functional abnormalities (e.g., insulin resistance), various symptoms (e.g., obstructive sleep apnea), and several comorbidity risks (e.g., cardiovascular diseases and type 2 diabetes) [3]. Similar statements have recently been made by other organizations, such as the Canadian Medical Association or the American Heart Association [4]. Such statements are rooted in a medical perspective on obesity, which contrasts with the positions of other organizations "who believe that defining obesity as a disease is unhelpful", such as the British Psychological Society [5]. The latter has emphasized social factors (e.g., weight stigma) and environmental factors, grouped under the wider notion of an 'obesogenic environment' (e.g., food deserts, neighborhoods that do not support walking) [5].

This tension echoes a deeper schism between a medical perspective emphasizing physical health and one that also considers mental well-being. Indeed, some groups view

weight loss as an 'obsession', noting that many individuals who attempt weight loss become heavier and their self-esteem shattered, while the general population may have a higher bias and stigma for those living with obesity [6]. Put simply, one side suggests that "success is reframed as improvements in health, function and quality of life" [6], whereas another side emphasizes weight loss as the core outcome of obesity treatment [7]. Opponents to a medical view on obesity include the Health at Every Size movement, which is weight neutral, thus rejecting weight as an indicator of health and opposing weight loss as an end goal [8]. These views resulted in today's body positivity movement [9] and proposals to characterize 'overweight' and 'obese' as stigmatizing and normative labels [10]. Collectives of feminists, fat activists, and health professionals similarly participate in 'anti-diet movements' [11]. Caught between different camps, policymakers are urged to reframe health policies [12], particularly as obesity policies have occasionally been designated as stigmatizing [13], and the ensuing weight bias has a demonstrated negative impact on mental health [12,14]. Thus, policymakers need to exert great care to satisfy different stakeholders when setting targets for population health [15], carefully consider the language within health policies (e.g., use of person-first versus disease-first terminology), and handle a complex problem driven by a massive number of factors.

In this paper, we use techniques from artificial intelligence (AI) and participatory modeling to support policymakers in balancing different perspectives on obesity and compare their implications for public policies. Our study is rooted in the growing "recognition that obesity management should be about improved health and well-being, and not just weight loss" [16]; hence, we integrate *both* perspectives in a comprehensive representation of knowledge on obesity. In particular, our study is motivated by a discussion paper from the Provincial Health Services Authority (PHSA) of British Columbia, in Canada. The paper, titled "From weight to well-being: time for a shift in paradigms?" acknowledged the long-established and well-documented links between obesity, the broader environmental context, and medical conditions [17]. At the same time, it suggested that shifting from weight-focused to well-being-focused approaches in practice and policy had the potential to improve population health and address the problems associated with obesity in ways that protect and promote mental well-being as well as physical well-being. Our study uses AI and participatory modeling to investigate this proposed paradigm shift and its implications for obesity policies.

AI has a rich history in obesity research [18]. Although it often evokes the use of machine learning to predict or diagnose obesity [19–22], other facets of AI underpin the foundations of numerous systems [23], such as the use of large language models (e.g., GPT) for personalized assessment [24] and treatment [25,26]. For instance, AI applications in smartphones support efficient ecological momentary assessments to collect precise data on human behavior, thus examining how individual and environmental characteristics shape eating and physical activity behaviors [27]. Serious games have also been used in obesity research [28], for instance, by incorporating elements of AI (e.g., recommender systems) to tailor interventions [29]. Simulations have also played an essential role in policymaking on obesity, for example through agent-based models [30,31], which are increasingly used in digital government research [32]. In an agent-based model, the AI components aim to replicate core aspects of human decision-making within virtual agents. In this paper, we focus on a traditional aspect of AI, which is to *represent and analyze knowledge* regarding a system. The Foresight Obesity Report [33] represented the system of obesity as a conceptual map, in which weight-related factors were captured as nodes and their interrelationship as edges. This representation has been adopted by many studies [34–37], often with a *participatory modeling* approach, whereby the knowledge embodied in the graph was obtained through human participants. Our work contributes to this literature in two ways:

- We create a concept map of obesity with 19 experts to combine different views on obesity, physical well-being, and mental well-being. With 98 nodes and 174 edges, it is currently the most comprehensive expert-driven model of obesity that conciliates medical perspectives with those emphasizing well-being.

- We analyze knowledge on obesity using tools from network science and natural language processing and examine the implications for public policymaking.

The remainder of this paper is structured as follows. To make this manuscript self-contained, we start in Section 2 with a brief introduction to the principles of participatory modeling; readers familiar with this field may skip the section. Based on these principles, Section 3 explains our methods to elicit knowledge in semi-structured interviews with experts and structure it into a conceptual model. Since the knowledge representation is only as strong as the experts who participate in this exercise, we provide the full list of participants for transparency in Appendix A. Our analysis of participants' knowledge is explained in Section 4, covering both an examination of the map using network science (Section 4.2) and the use of natural language processing over the transcribed interviews (Section 4.3). Since the map is a massive model spanning multiple domains (e.g., built environment, psychology, physiology), space limitations preclude its inclusion in the main body of this paper; it is instead detailed in Appendix B. Finally, we discuss the implications for policymaking in obesity and potential extensions so that improved AI solutions can further support government services and policies.

## 2. Background: A Primer on Participatory Modeling for Causal Mapping

*Participatory Modeling* (PM) is an approach that actively identifies and engages participants in knowledge elicitation and integration. In contrast to other fields of participatory research (e.g., citizen science), PM tends to focus on motivating changes and is considered a researcher-driven approach to taking action [38]. Core application areas of PM include policy and planning [39]. Although there is a wide variety of tools and methods for PM [40], many of them represent a conceptual system in schematic form. The schema may be loosely structured (e.g., rich pictures) or follow a prescribed structure (for a comparison on the level and nature of the constraints, we refer the reader to Table 5 in [41]) such as a map (e.g., cognitive map, causal loop diagram) [42–46], in which case, the approach is specifically known as 'participatory systems mapping' [47] and serves to assist with public policy decisions in contrast with group-based methods that may seek to empower communities (e.g., community-based systems modeling) [48]. Note that participatory systems mapping is distinct from 'participatory mapping' [49], which is a term used for collecting volunteered geographic information; they share the notion of contributions from participants and an application to participatory planning and decision-making, but participatory mapping focuses on spatially referenced data and involves geographic information systems.

In this paper, we represent knowledge in the form of a *causal map*, where weight-related factors are represented by nodes and their relationships are captured by edges. Any factor can be represented, but it must be clearly able to increase or decrease; for example, 'weather' or 'built environment' would be poor choices for factors, but 'rain level' or 'availability of sidewalks' could be used. A relationship denotes causality; hence, it is directed and typed (either positive or negative) to indicate that an increase in a node either increases ('+') or decreases ('−') another node. For instance, *cardiovascular diseases* $\xrightarrow{+}$ *fear of engaging in physical activity* indicates that individuals with cardiovascular diseases have a higher risk of fearing physical activity. Conversely, *fear of engaging in physical activity* $\xrightarrow{-}$ *physical activity* states that the higher the fear of engagement, the lower the level of physical activity. Hasty conclusions should be avoided on the basis of a single relationship: for example, we *cannot* conclude that all individuals with a fear of engagement will automatically have a low level of physical activity. Rather, a map represents knowledge regarding an entire system; hence, we need to consider the joint effects of all protective and risk factors [50] (e.g., the fear of engagement may be counterbalanced by a protective factor for physical activity). Note that there are variations of this process [47] in which types include 'unclear' (we know that there is a relationship but its type is unknown) or 'complex' (when it is non-linear or depends on other considerations). For example, weight-based discrimination may not exist at certain levels of weight but rises rapidly from a certain level onward, which shows a non-linear relationship that would only be typed as '+' under certain conditions [51].

Such variations are not used in the present work, where edges are strictly categorized as increasing or decreasing.

There are several core steps to arrive at a causal map through a PM process. The specific number of core steps can differ across papers depending on the granularity chosen by the authors. For example, some may split the elicitation stage into a development phase (e.g., pilot testing) and its application with participants [52]. Despite differences, practitioners generally agree on two aspects of the steps. The *sequence* starts with ① a definition of the objective and scope, followed by the ② stakeholder selection, ③ knowledge generation, and ④ qualitative aggregation into a map [53]. These steps account for four facets, known as the four Ps [54]: purpose (why this model and a PM approach?), partnership (who participates?), process (how are they involved?), product (what is the result?).

To begin, ① setting a focal point and a problem boundary contributes to operationalizing the purpose of the model. The focal point can consist of a single variable (e.g., obesity [51], smart city [55]) of a set of variables (e.g., suicide ideation, attempt, and death [50]). *Before* the creation of a representation of a system, the boundaries provide high-level guidance to exclude or include certain concepts. For example, a model for adult obesity should not include childhood obesity. *After* the system is represented as a map, its boundaries become more clearly delineated by its actual content. That is, "the boundary of such a conceptual system is made of the concepts contained/described within it." [56]. Although a study usually starts by setting high-level boundaries, there are exceptions in which boundaries are only set after interviews have been transcribed [57].

To ② select stakeholders, the overarching principle is that "researchers purposefully select participant stakeholders based on expertise or vested interests in the research" [39]. Most studies systematically identify the knowledge areas needed to solve the problem [58]. The corresponding identification of participants is rarely systematic in PM studies and involves a variety of methods, such as sampling (which may include referrals for snowball sampling), nomination by a committee, or the application of selection criteria (e.g., who qualifies as an 'expert'?) [58]. Although selecting stakeholders is primarily a matter of partnership, it also starts to venture into the process. For instance, when participants are invited, an email can explain the process to them. This helps to set the tone and 'manage expectations', which is one of several characteristics of successful PM projects [59].

The ③ knowledge generation step is often the most detailed in studies and also the most time-consuming since it is process-intensive. Participants could be engaged individually via semi-structured interviews (e.g., in person or remotely, through a facilitator or a device [60]) or as a group through workshop activities such as writing factors on post-it notes, consolidating them and linking them collaboratively [47]. Each approach needs a specific skill set from the facilitator; for instance, workshops require paying particular attention to social and group dynamics such that there is equity among participants. The nature of the engagement is important because the co-production of knowledge at this step can be a strong benefit of a PM approach. Indeed, knowledge co-production contributes to greater trust in the model [61], and endorsements of a model by community champions and/or leading experts can contribute to an efficient translation from knowledge to action. Finally, we arrive at an ④ aggregate map, which invokes both a product and a process (how are participants involved in validating and using the map?). The map may have been directly constructed from a group in a workshop or will result from combining the maps of individuals. Neither approach is easier than the other; rather, they depend on the needs of the project and the skills of the team. For instance, a workshop activity may require an expert facilitator to promote equity and avoid conflicts. In contrast, individual interviews may be simpler, but combining individual maps requires either accounting for the heterogeneity of the terminology used by participants [62], or limiting it by only allowing for certain pre-selected terms [63]. The aggregate map may also be augmented based on the literature and data [64,65].

## 3. Knowledge Representation by Participatory Systems Mapping

### 3.1. Key Steps of Our Methodology

The four stages of the methodology (①–④) refer to the steps explained in the previous section. Our methods are summarized in Figure 1 and detailed below.

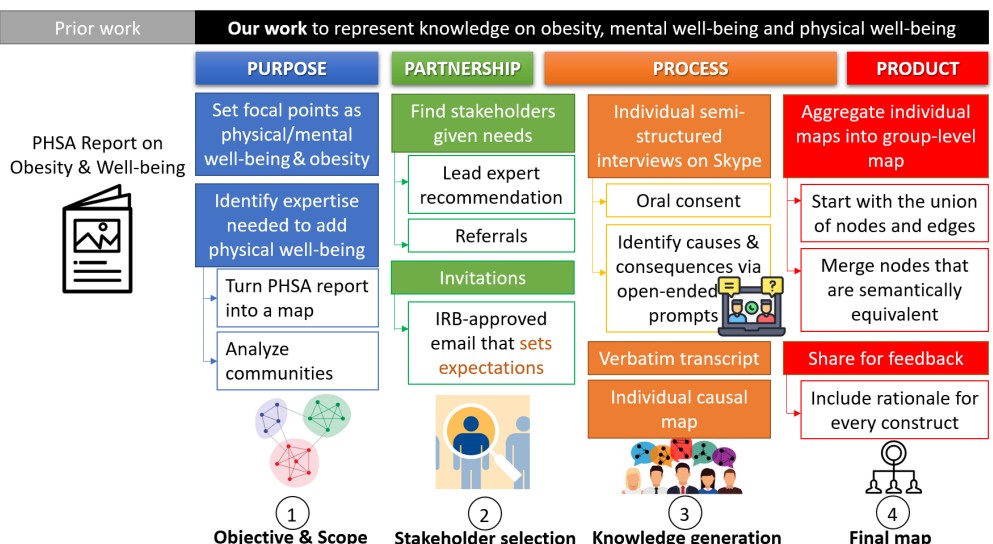

**Figure 1.** The four main stages of our process (objective and scope, stakeholder selection, knowledge generation, and the final map) each involve a set of steps and sub-steps. The PM literature also outlines methodological choices based on the four P's [54] (purpose, partnership, process, product), which partly align with these stages.

Three elements constitute our focal point ①: physical well-being, mental well-being, and obesity. We focus on adults within Canada; hence, childhood obesity is excluded from our map. The relationships between mental well-being and obesity are detailed in a report by the Provincial Health Services Authority (PHSA) of British Columbia, so we use their report as a *point of departure*. Specifically, we transform their report into a map and analyze it to unveil areas that are already well-covered and identify those needing expert input. This is similar to the approach of Moon and Browne, who disaggregated their complex system "into smaller subsystem modules that each represented a functioning unit, about which an individual is likely to have more comprehensive knowledge" [52]. Note that the notion of 'subsystem modules' or 'sub-models' is common in participatory systems mapping but these components are sometimes found *after* individual maps are combined [66]. Identifying the components *before* the interviews helps us distribute the knowledge elicitation across participants and ensure that we have sufficient participants for each facet of the problem.

Given the desired areas of expertise, we select stakeholders ② who can cover these areas based on either significant research expertise or relevant experience in policymaking. We do not require that participants live in Canada, particularly when it comes to medical expertise (e.g., how inflammation or cardiovascular diseases are caused). We do require knowledge of the Canadian setting for constructs that are place-specific, such as the built environment (e.g., the existence of food deserts). Stakeholders were primarily identified by a leading researcher (see acknowledgments) with additional referrals; this is the same approach as in past studies employing purposive sampling facilitated by the positioning of a researcher, supplemented by snowball sampling [67]. There is no 'gold number' on how many individuals should be invited or ultimately interviewed. For example, recent PM studies have interviewed 7 researchers [68], 11 [69] or 14 stakeholders [66], 15 subject-matter experts [70], or as many as 30 individuals [71]. With 19 participants, our project is on the higher end of the distribution for PM projects performing individual interviews.

All experts contacted agreed to participate and provide their names for full transparency. As shown in Appendix A, participants provided a breadth of expertise and significant experience on topics that directly contribute to our core topics. For example, physical well-being was addressed by our multiple experts on physical activity, as well as experts in human physiology. Mental well-being was covered through expertise in domains such as social determinants or healthy living. Per the terms of the IRB approval granted by Simon Fraser University (approved by the Office of Research Ethics under the application titled 'From weight to well-being: assessing drivers and mechanisms'), we cannot associate the identity of a participant with any specific element (e.g., cannot attribute a quote or construct in the map). Every participant received the same IRB-approved email invitation for a Skype interview, including study details and the consent form. Participant interviews were recorded on Skype, including the provision of oral consent.

We then conducted individual semi-structured interviews ③. Although a group setting would "enhance mutual learning and problem-solving" [53], mutual learning is not the purpose of our work since participants are selected because they are already experts with aspects of the obesity system. Rather than a social objective, our project was designed to support decision-making, policymaking, and management of the obesity system. In addition, PM research has noted that conflict across shareholders is a potential challenge [59]. Interviewing experts individually is part of how we avoid this conflict, since "one-on-one interviews also allow participants to express views they might be hesitant to express in a group setting" [39]. Each semi-structured interview is performed as detailed in [50] by teasing out causes and consequences. As noted by Sterling and colleagues when extracting a conceptual model from discussions, "modelers must strive to ensure their own views and favored methods do not drive the model development" [59]. Consequently, the interviewer avoids confirmatory questions that may plant a concept in the interviewee's mind (e.g., 'do you think that body image increases well-being?'), and instead uses open-ended questions (e.g., 'what do you think contributes to increasing well-being?') and summaries (e.g., 'you stated that body image and physical activity increase well-being; is there anything else?'). The interviewer does not judge the participant and does not distract the participant by showing a map or notes. In line with previous research, "these interviews were then transcribed and coded for key terms. The relationships among the key terms elicited were then visually represented (by the research team) as a concept map depicting the causal linkages mentioned by the interviewee." [72]. The individual maps resulting from semi-structured interviews are finally ④ aggregated into a group-level map, which accounts for the variation in terminology. Since the participants provide complementary perspectives on different facets of obesity and well-being, we cannot 'select' one participant as a base map and add the others; rather, we focus on integrating them and preserving information. When this group-level map is ready, we document it in a report that includes a rationale for every node and edge and share the report with all participants for feedback and corrections. Soliciting feedback is also part of how we navigate the potential for conflict across stakeholders, as experts are then made aware of the content originating from other participants. Our process of sharing the map *along with* explanations for every construct is similar to the process by Swierad et al., in which the map was shared for review together with contextual narratives [73]; this goes beyond typical practices for PM studies, in which participants provide feedback by reviewing only a printed version of the map [74].

### 3.2. A Starting Point: The PHSA Report on Obesity and Well-Being

We manually coded all the relationships mentioned in the PHSA report. Each relationship was directional: a change in one factor triggers a change in another factor. For example, the sentence "bias and stigma have significant negative consequences, including overeating and avoidance of exercise and psychological harm" led to coding "weight stigma impacts overeating", "weight stigma impacts exercise", and "weight stigma impacts psychological harm". Relationships that endorsed associations rather than causations were accounted for as two-directional relationships. For example, the sentence "severe obesity is associated

with a lifetime history of binge eating" was coded as "obesity impacts binge eating" and "binge eating impacts obesity". The set of all relationships constitutes a map, whose characteristics were explored using visual analytics. The maps are provided in Figures 2–4, where each circle stands for a concept, and causations are depicted via curves (i.e., the presence of a curve linking two factors means that one causes the other). When the report provided association rather than causation, the map shows two curves denoting causations running both ways. The fact that relationships overwhelmingly depicted causations rather than associations can be seen in the figures, as two factors tend to be connected by a single curve rather than two curves. Note that these maps only aim to depict relationships; they do not state whether the relationships are positive, negative, or part of feedback loops.

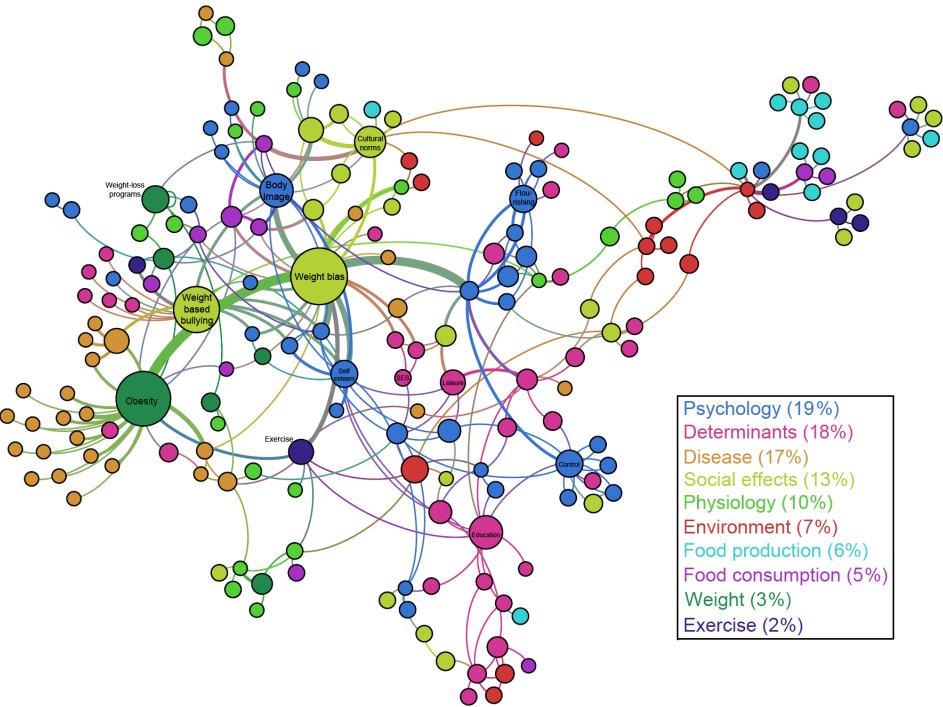

**Figure 2.** Analysis using the Foresight categories. Social *determinants* cover concepts such as smoking, socioeconomic status, educational attainments, employment inequities, work-life balance, and food literacy. Social *effects* include notions such as weight-based bullying, weight bias, and stigma.

In the analysis stage, we first assigned a category to each factor based on the categories used in the Foresight Obesity Report [33]. Coding new maps against the thematic clusters defined in the Foresight Obesity Report has indeed become common practice in PM studies on obesity [75]. In addition to the categories from the Foresight Obesity Report, we created the "Disease" category to account for the fact that numerous comorbidities were mentioned in the PHSA report. There are three main findings from this first analysis (Figure 2). First, it highlights that the PHSA report focused on the social and psychological contributors to obesity and well-being; while numerous comorbidities were cited, they mostly came from one subsection of the report. In comparison, physical activity, and exercise account for relatively few factors, as do food production and consumption. Second, using the categories from the Foresight Obesity Map does not result in coherent groupings of concepts (i.e., clusters); the map in Figure 2 cannot be straightforwardly divided into components based on the color of the nodes. This suggested that more specific categories needed to be developed in order to adequately capture the uniqueness of the PHSA report. Finally, this analysis found that the comorbidities (in brown on the left of Figure 2) were only listed as consequences of obesity and did not influence other factors. This is a major gap in the PHSA report given the wealth of evidence regarding the impact of comorbidities on well-being. For example, the PHSA report accounted for the association between obesity

and high blood pressure (i.e., hypertension). However, it did not account for the multiple studies showing the impact of hypertension on people's well-being, for example, in terms of health-related quality of life [76].

Having noticed that categories from the Foresight Obesity report lead to scattered coverage, the second phase of our analysis searched for categories tailored to the PHSA map. This is similar to previous analyses of systems maps, in which the same system was decomposed into different sets of categories to understand its dynamics in complementary ways [57]. To automatically find categories that fit the PHSA map, we assessed which modules were present in the map. A module is a list of factors that are 'structurally coherent'; that is, these factors are mathematically more linked to one another than they are to factors in other modules. The results (Figure 3) suggest seven categories that better identify the specificities of the PHSA report. In particular, this points out the strengths of the report regarding its thorough examination of eating disorders and weight stigma. This analysis also uncovers new gaps in the report: the fact that concepts linked to nutrition are relatively isolated indicated the need to bring in more clinical evidence about how nutrition mediates the relationships between factors.

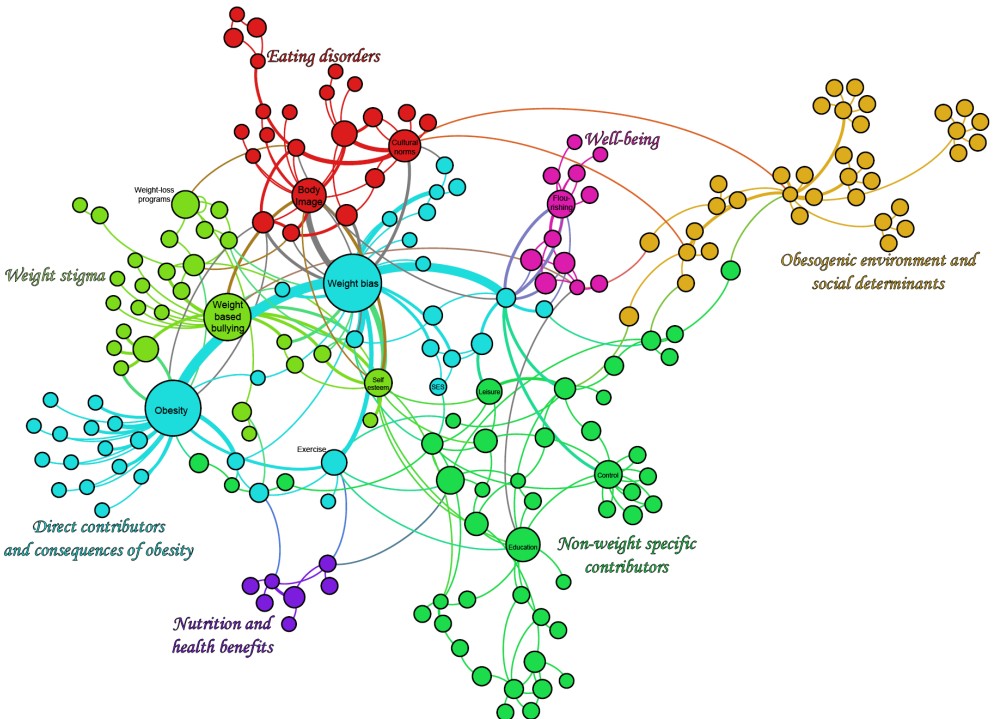

**Figure 3.** Analysis using categories specifically designed for the PHSA report. Each color corresponds to a category: for example, nodes and edges involved in eating disorders are shown in red.

In the final phase, we examined which concepts played an important role in the map. This was conducted by considering two metrics: the degree, which is the number of relationships that a factor is involved in, and betweenness centrality, which is the extent to which a factor ends up on pathways between other factors. Degree and betweenness are the two node centrality measures commonly applied in the analysis of obesity maps [35]. In particular, previous work found a correlation ($R^2 = 0.51$) between degree centrality and body mass index [77]. The results show that obesity is the most central, followed by themes pertaining primarily to mental well-being (e.g., weight bias, weight bullying, self-esteem, body image); the prominence of such constructs and the lack of precise physiological or environmental notions reinforce the findings from the previous two analyses.

To summarize, our assessment of the PHSA report has helped clarify the strengths on which our own map can build, as well as the areas that need to be expanded. Both aspects are listed in Table 1. Given our primary objective of bringing in solid clinical evidence,

our analysis shows that we need to build on the PHSA report in three areas: clinical pathways (e.g., to capture the impact of comorbidities on obesity), physical well-being (e.g., to complement the previous report's emphasis on mental well-being), and resources that enable a high level of physical well-being (e.g., the environment in which individuals live). While genetics are clearly involved in understanding the clinical components of obesity, they were not proposed as one of the key themes due to the context of supporting efficient policies for well-being. While food production and consumption are heavily influenced by policies, we continue to underemphasize these two components given our focus on physical well-being as it relates to overweight and obesity.

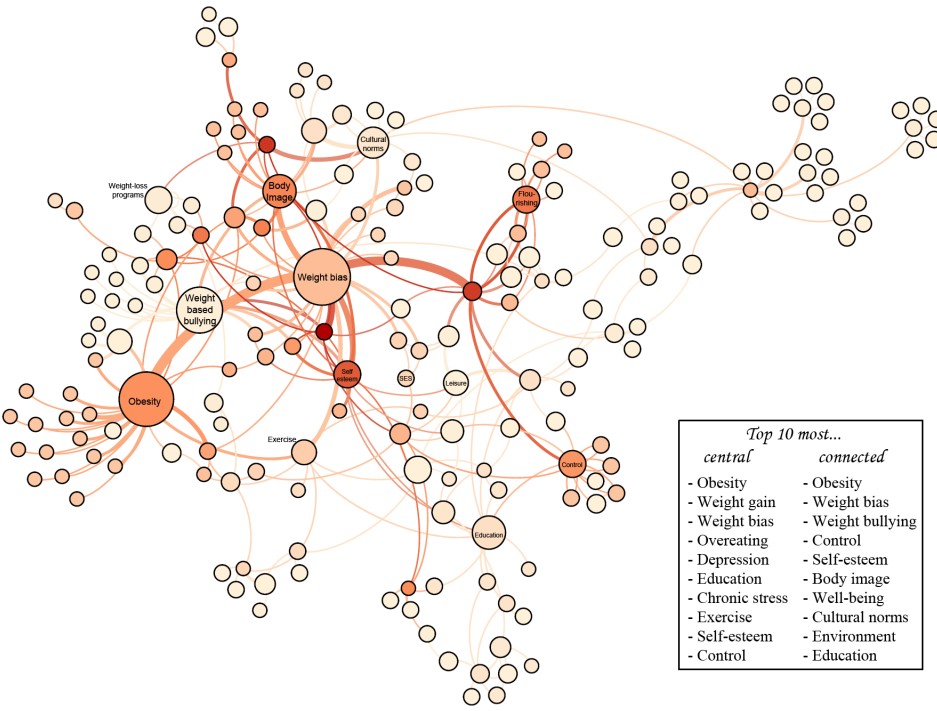

**Figure 4.** Comparison of the most important factors by connection (shown by circle size) and centrality (shown by hue).

**Table 1.** Strengths of the previous report and areas emphasized/peripheral in our work.

| Strengths of the PHSA Report | Areas of Emphasis in Our Work | Areas Peripheral to Our Work |
|---|---|---|
| Psycho-social pathways (e.g., consequences of weight stigma) | Clinical pathways (e.g., consequences of comorbidities, impact of nutrition) | Food production |
| Mental well-being | Physical well-being | Food consumption |
| Resources impacted by obesity (e.g., job opportunities) | Resources enabling a high level of physical well-being (e.g., the built environment) | Genetics |

### 3.3. Extending the Map via Semi-Structured Interviews

As detailed in Section 3.1, our identification of weaknesses and strengths served to set the agenda for identifying participants. We initially identified a total of 54 experts based on recommendations from a leading expert for domains that need to be covered. In addition, 6 experts were identified through referrals. We did not invite all experts at once; rather, we gradually invited experts in each domain until sufficient coverage was obtained. Out of the 60 experts identified, we invited 26 based on our specific needs throughout the process. A total of 22 individuals agreed to participate (i.e., 4 did not return our email), out of whom, 19 set up an appointment time and were interviewed (i.e., 3 agreed to participate but did not return emails to set up a time). All interviewees are listed in Table A1 (Appendix A) with their permission.

A student facilitator was hired for this project. After the first round of selecting applicants based on grades and recommendation letters, applicants were evaluated based on the same set of project-specific questions, which included domain knowledge (e.g., 'Can you cite three social determinants of obesity?') and situations (e.g., 'Imagine that you have to interview one of the most renowned experts of obesity world-wide. How would you prepare?'). Once hired, the facilitator was trained on running semi-structured interviews and practiced on three individuals, who were not among participants but had a sufficient level of awareness on the application field as Ph.D. holders and published scholars in obesity research. After training, semi-structured interviews with participants began. An appointment at a time of their convenience was set for an hour. In line with previous guidance on semi-structured interviews for causal mapping [57], an interview "continues until a sufficient level of depth is reached or until the insights of the interviewee are exhausted, at which point the chain of causality may be built up further during interviews with other participants". On average, each recorded interview lasted for 40 min, with most taking 27 to 37 min and a few lasting an hour. These durations are similar to those encountered in other works on mapping complex systems, such as suicide [70].

We employed a professional transcription service to turn each recording into a text document. On average, one transcript contained $4592 \pm 1452$ words, leading to a total of almost 90,000 words. We manually extracted a map from each transcribed interview. Although there is emerging technology to automatically derive maps from text [78], PM studies primarily use software to help with the manual process of annotating (e.g., nVivo) rather than replacing it entirely with algorithms. We used a dedicated setup to manually combine the maps into one (Figure 5). In this case, emerging technology for combining maps does not offer the same level of accuracy as manual processing; hence, it remains the gold standard [62]. The contribution of participants to the combined map reflects their areas of expertise. For example, an expert on health literacy and health inequities would not contribute to identifying and linking physiological constructs such as adipose tissue metabolism and lipoproteins. As expected, there are substantial differences regarding the *parts* of the map that are shaped by individual experts.

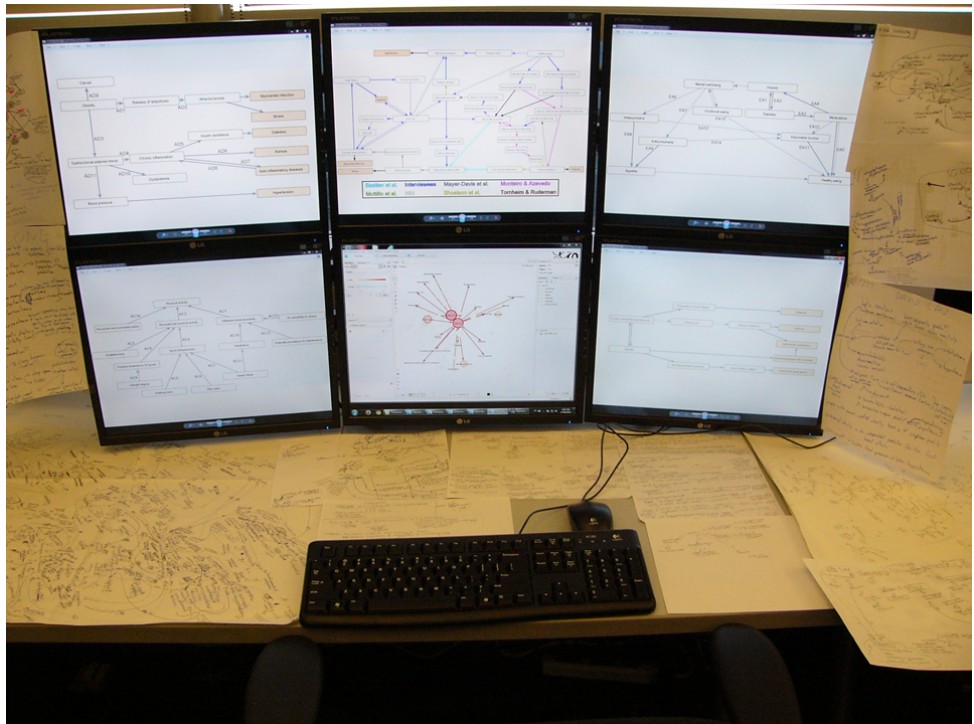

**Figure 5.** Set up to identify similar concepts across individual maps and combine them. *Low resolution is used to avoid the association of participants with specific statements while illustrating our overall set-up.*

## 4. Analyzing Knowledge on Obesity and Well-Being

### 4.1. Overview of the Map

Our final map is available on the third-party repository of the Open Science Framework [79] at https://osf.io/7ztwu/, under 'sample maps'. Our map is named Giabbanelli & Macewan; other maps on obesity are provided for comparison and include the Foresight Obesity Map as well as the work of Verigin et al. [80]. A criticism of participatory modeling approaches in health is that "model assumptions and structures tend not to be opened up to scrutiny" [81]. Given this specific concern, the assumptions and structure of the many components of the map are detailed in Appendix B; this appendix was also shared with participants as part of gathering feedback. The map consists of 98 concept nodes and 174 directed–typed relationships. The size of our map is about average for a systems map related to obesity, between smaller maps obtained for childhood obesity in South Africa (47 factors) [74] or Manhattan's Chinatown (39 factors) [73], and larger maps such as the Foresight map (108 factors) [33] of the work of McGlashan et al. (114 factors) [75].

This section focuses on the analysis of the map. Although network analysis (as shown for the PHSA report in Section 3.2) is the common approach employed in systems maps [82], our work goes one step further by also employing natural language processing to examine the transcribed audio recordings of stakeholder interviews. A manual content analysis of the interviews is fairly common in PM studies [41]. In studies, such as the study by Radonic or LaMere et al., recordings served as a form of indirect elicitation to verify that the map had the right content and level of details [83,84]. In other works, such as Kropf et al. [85], an aggregated map was constructed from a workshop, and *then* semi-structured interviews were performed, with content analysis informing refinements of the aggregated map. Our use of NLP has similar objectives but promotes a computational approach that scales up with the data volume of almost 90,000 words.

### 4.2. Network Analysis of the Conceptual Map

We used the conceptual map to compare the perspectives around well-being with those around obesity (Figure 6). We found that different sets of factors were involved in these two perspectives (Table 2). Two specific differences were noted between these perspectives. First, obesity evokes a *medical* focus on *weight as a problem*, whereas well-being leads to a more *solution-focused* approach that is *broad* and diverse.

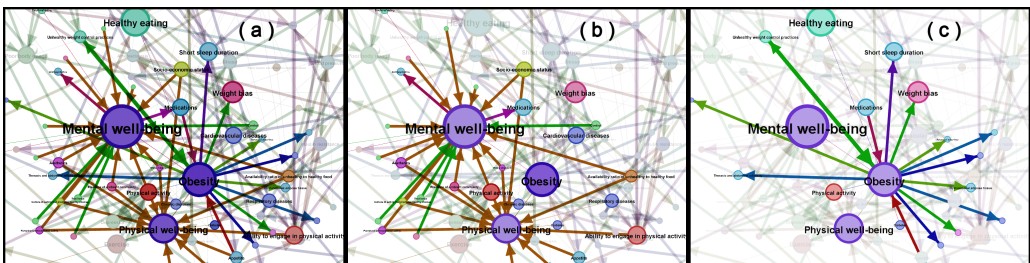

**Figure 6.** A systems view of obesity and well-being taken together (**a**), compared to being centered on either well-being (**b**) or obesity (**c**). Colors indicate that nodes belong to different categories.

We performed an analysis to check these findings by creating a *reduced version* of our conceptual map (Figure 7). The idea of reducing a map of obesity based on its communities was pioneered by Finegood [86] and then used in studies by other scholars [75]. Examining the map at a higher level of abstraction allows us to observe patterns between themes, which may not be apparent when working at the level of specific concepts. Indeed, "there is growing evidence of the effectiveness of applying multilevel network methods in diverse fields to identify meaningful patterns of interdependencies across multiple levels of a system, relative to what can be understood via simplified, single layer network analyses." [87]. To obtain a reduced version, factors were gathered by categories, and a relationship existed between two categories if and only if there was a relationship between

factors in these categories. For example, a relationship existed in our conceptual map between socioeconomic status and mental well-being; hence, there is a relationship in the reduced map between the categories of determinants and well-being. Note that there is no gold standard on the number of categories that should be used. From a small number of PM studies using reduced maps, it appears that the number of categories may scale with the number of factors. A map of the obesogenic environment for children in Pennsylvania had 3 clusters for 32 features [77], one on obesity in Manhattan's Chinatown had 4 themes for 39 factors [73], and another in South Africa had 5 domains for 47 factors [74]. In a different domain, a map for HIV in sub-Saharan Africa had 4 or 5 domains for 71 nodes. Since our map is larger, we decomposed it into more domains accordingly.

**Table 2.** Examples of factors that connect to either obesity or well-being.

|  |  | Obesity | Well-Being |
| --- | --- | --- | --- |
| Causes |  | Medications | Perceived environmental safety |
|  |  | Overeating | Presence of a vibrant community |
|  |  | Physical activity | Resilience |
|  |  | Diabetes | Ability to engage in physical activity |
| Consequences |  | Short sleep duration | Antipsychotics |
|  |  | Cancer | Medications |
|  |  | Dysfunctional adipose tissue |  |
|  |  | Weight bias |  |

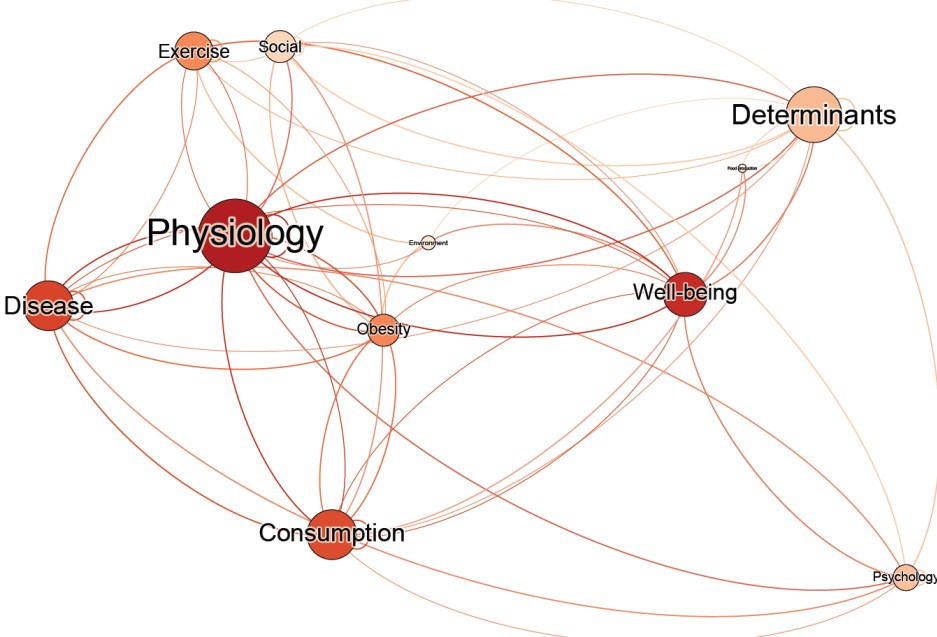

**Figure 7.** Reduced conceptual map using categories. The size of each category indicates its degrees (i.e., number of interactions) while its color is indicative of centrality.

This reduced map shows that well-being is directly connected to all other categories. From an obesity-centric perspective, food production, and social factors are more indirect and distal. That is, obesity is certainly impacted by food production but this was viewed through the lens of food consumption. Consequently, this reduced map confirms the focused nature of an obesity-centric perspective compared to the more diverse themes produced by a well-being-centric perspective.

### 4.3. Natural Language Processing for the Interviews

We analyzed the interviews to compare the language used when discussing obesity compared to well-being. The unit of analysis was at the level of a respondent's answer. That is, each interview was divided into a set of answers. These answers were then categorized into two sets: answers that included obesity and answers that included well-being. Note that these sets are not exclusive since an answer may speak of both. In these two sets, standard text analysis procedures were used to transform the words. Specifically, words were stemmed (e.g., "problems" and "problem" are seen as the same) and common English words (e.g., "and", "or", "that") were removed. Then, we counted the frequency of each word in each set. Comparing the frequencies (Figures 8 and 9) reiterates the finding that obesity emphasizes the problem from a medical perspective, whereas well-being triggers a more diverse solution-oriented language.

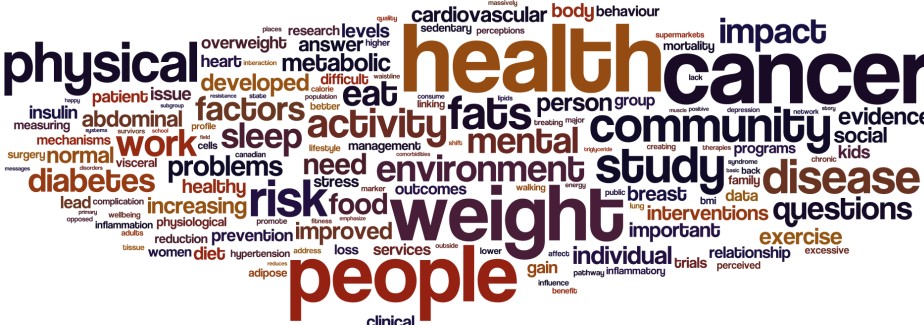

**Figure 8.** Word clouds for answers including obesity.

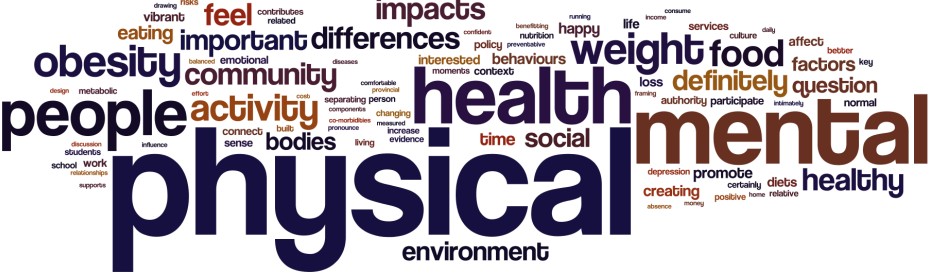

**Figure 9.** Word clouds for answers including well-being.

## 5. Discussion and Future Work

### 5.1. Structuring the Relations between Physical and Well-Being in the Context of Obesity

The representation and analysis of knowledge are fundamental themes within artificial intelligence (AI). In this paper, we used AI through the lens of participatory modeling (PM), network analysis, and natural language processing (NLP) to examine the tension between different views on obesity and well-being. We started with a previous report as a point of departure, where mental well-being was emphasized. Our work disentangles the different components of well-being by accounting for both physical and mental well-being. Interviewees overwhelmingly reported that these two facets are closely interrelated:

> "You don't have two separate drawers, one is mental health, one is physical health. They're really part of the same thing."

However, most of the interviewees hinted at a *temporal separation* between these two concepts. Factors can first influence physical health, which in turn impacts mental health, or vice versa. Table 3 summarizes the factors that interviewees thought would influence physical or mental well-being first, and the factors that would influence both at the same time. When examining how mental and physical well-being affect each other, interviewees were more likely to provide examples of mental well-being influencing physical

well-being than the other way around. This may be due to a cognitive bias: if there are several intermediate factors mediating the impact of physical well-being on mental well-being, then this impact may feel indirect and interviewees can be less aware of its existence. It is also possible that framing the discussion around the components of well-being may have influenced interviewees' responses toward mental well-being since physical well-being is commonly referred to as physical health. Furthermore, prompting interviewees to think about the differences between physical and mental well-being could have biased their answers toward an assumption that there is a difference. However, this does not seem to be the case. For example, an interviewee began by saying that "the difference is huge" but, like all other interviewees, eventually concluded that the two are interrelated: "I believe that they can't go, they shouldn't be looked in isolation".

Two interviewees raised the importance of including other dimensions of well-being. In particular, they raised the need to take into account *social well-being*. However, they defined this concept differently. One defined social well-being as having the resources to play a role in society:

> "That would entail having access to the resources that are required for everyday living. Access to shelter, to clothing, to reasonable foodstuffs, to entertainment, meet entertainment needs so that people could participate meaningfully in the society or the culture in which they find themselves".

The other interviewee viewed social well-being more in terms of social capital, in the context of close family ties or community ties. Overall, this suggests that future studies could expand on well-being by going beyond the physical and mental aspects that were considered here.

**Table 3.** Factors that affect mental and physical well-being (listed by interviewees).

| Mental Well-Being | Physical Well-Being | Both |
| --- | --- | --- |
| Bias, bullying, culture of eating that promotes healthy choices instead of weight loss, deep relationships, discrimination, feeling able to contribute, feeling comfortable, feeling valued, happiness, physical activity, presence of a vibrant community, psychological stability, self-confidence, stigma | Ability to be physically active, appetite, broken bones, built environment, cardiovascular health, diabetes risk, eating behavior, energy balance, food quality, hormonal systems, metabolic health, nutrition, overweight/obesity, physical activity, sleep | Availability of healthy food options, built environment (pollution level, aesthetics, infrastructure), community ties, the culture of eating healthy food, eating behavior, economy, exercise, family (where kids go, how they arrive there, what they eat, how they spend their time), income, perceptions of neighborhood safety, political climate, public health messaging, school environment, work environment |

### 5.2. Limitations and Suggestions for Future Studies

Our participants were chosen for their expertise either in obesity research or in obesity policymaking, and participants were gradually added to the study to cover domains that had to be expanded from previous works. Despite these strengths, every study that uses a participatory modeling approach to synthesize knowledge on a subject is necessarily limited to reflecting the perspectives of its participants. Complementary perspectives would be offered by directly interviewing individuals with different lived experiences of overweight and obesity. Previous studies have indeed shown that the mental models of individuals directly concerned with a problem could differ from those who are experts in the management of this problem, thereby resulting in different maps [88]. Our previous studies further showed that individuals have very heterogeneous perspectives on obesity [89]; hence, accounting for lived experiences is unlikely to result in just 'an expert map' and a single 'map of lived experiences'; rather, *several* maps may be produced by identifying shared features [63] or trajectories among lived experiences.

Our network analysis was guided by the focus of our research question on a proposed paradigm shift from obesity to well-being. The publicly released map that we have constructed could be analyzed in numerous other ways, in response to other specific questions. For instance, network analysis often serves to identify "critical components within an interconnected system" [90], thus providing insight for public health in terms of setting intervention targets. Thus, our map could be analyzed with respect to several

node centrality algorithms since hundreds of them are now available and applicable to sparse networks such as systems maps [91]. There have also been manuscripts dedicated to finding feedback loops in an obesity map [67], or identifying data gaps in obesity research [92]. Using the analysis of a map as a proxy for designing interventions on a complex system has been a growing trend, as scholars noted in 2019 that these "whole-of-system interventions represent the next step in the science of obesity prevention" [87]. While our focus is on obesity and well-being, we note that obesity research is an interdisciplinary field and similar mapping efforts have been undertaken in related areas such as physical activity or nutrition. Indeed, there is also growing literature in nutrition that conceptualizes and discusses systems components or use-system-oriented frameworks to assist stakeholders with design and evaluation [93]. Such works illustrate that the practice in PM and obesity has often been to detail the process of knowledge representation and publicly release a map (as we have done here), such that other scholars can benefit from it.

Similarly, our use of natural language processing sought to shine a light on the differences between the language used in obesity and well-being-centric perspectives. A variety of other NLP analyses are available and we have employed them elsewhere [94], such as sentiment analysis, topic modeling, or sarcasm detection. Many of these analyses have been conducted regarding obesity in social media, for instance, to study stigma in public opinion [95] or the relationships between health and place [96]. Since our expert sample would not be suitable for studies of public opinion or geography, these NLP techniques would not be *directly* applicable here. However, future works may use the content of our map to guide data collection efforts in social media, thus revealing how the general population thinks about various facets of the obesity system [97] and how interventions may be framed for public communication [98].

Through the interviews, experts expressed that relationships differed in strength. For example, it was said that pain is "certainly one of the most important" biomedical barriers to physical activity, or that having harmonious relationships with neighbors "would be really important" in shaping people's perceptions of their neighborhood. While the map presented here only articulates the relationships between numerous factors, including the strength of these relationships would allow practitioners to utilize the map for "what-if" scenarios. For example, practitioners could ask what would happen if an intervention were to be implemented on weight bias: the weighted map would then show how changes in weight bias would diffuse through the whole system. Consequently, a weighted map could serve as a decision support tool that is particularly valuable when facing the complexity of a phenomenon such as obesity and well-being. Previous endeavors have produced weighted maps for obesity [51,80], but they have always been smaller (i.e., fewer factors); hence, there is still a need for a systems map that precisely characterizes each relationship.

### 5.3. Implications of the Map for AI Solutions Focused on Well-Being

Numerous solutions seek to leverage advances in AI to improve well-being in the context of obesity [18]. Although the goals of these solutions may vary (e.g., prevention vs. treatment), many of them focus on constructs that are *proximal* to individuals [19–22]. This emphasis on individual responsibility is perhaps expected for self-management tools, and it is also found in other complex health problems [50]. However, the social–ecological framework has made it clear that well-being is *not* only a matter of individual determinants, it is also about how individuals engage with their peers, the communities, and environments in which they live, and the norms conveyed in society. Few AI solutions recognize that individual behaviors are shaped by their environment [27]. In this context, our map can help to avoid an exceedingly reductionist perspective. By *serving as a knowledge repository*, the map can, thus, guide the design of new tools. For example, apps for self-management can continue with asking questions about the individual (e.g., patterns pertaining to eating and exercising, sleeping patterns, self-esteem), and they can now follow up on such questions either within a level (e.g., to identify forms of exercise that are appropriate for an individual's capacity and improve their health) or across levels (e.g., to assess whether low

self-esteem is the product of a broader issue in interactions with peers or societal norms). The potential for the map to improve knowledge representation and reasoning related to well-being and obesity is also valuable for tools building on emerging technologies, such as large language models [24–26]. As shown in recent case studies [99], models such as GPT do not inherently encode causality related to obesity; hence, they need to be fine-tuned. The content of our map can, thus, fine-tune GPT, providing it with the knowledge of domain experts before using it to guide individuals in a holistic manner.

## 6. Conclusions

Obesity-centric and well-being-centric perspectives offer two different paradigms through which to frame public policy interventions. We created and analyzed a systems map of obesity and well-being, as well as analyzed key-informant interviews with experts in the field. We found several key differences between focusing on obesity versus focusing on well-being. A well-being-centric perspective tended to be more comprehensive and included all factors identified by experts as pertinent to the topic, whereas an obesity-centric perspective was less connected to several factors, notably food production and social factors. Furthermore, discussions of the well-being-centric perspective tended to generate solution-focused language, whereas the obesity-centric perspective generated problem-focused language. In conclusion, this research demonstrated that the paradigm through which weight is viewed can impact which factors are considered relevant to the problem and the type of solutions that may be proposed.

**Author Contributions:** Conceptualization, P.J.G.; methodology, P.J.G.; formal analysis, G.M.; investigation, P.J.G. and G.M.; resources, P.J.G.; data curation, P.J.G.; writing—original draft preparation, P.J.G. and G.M.; visualization, G.M.; supervision, P.J.G.; project administration, P.J.G.; funding acquisition, P.J.G. All authors have read and agreed to the published version of the manuscript.

**Funding:** This research was funded by the Provincial Health Services Authority (PHSA) of British Columbia. There is no grant number.

**Institutional Review Board Statement:** The study was conducted in accordance with the Declaration of Helsinki, and approved by the Institutional Review Board (or Ethics Committee) of Simon Fraser University (protocol 'From weight to well-being: assessing drivers and mechanisms', February 2014) for studies involving humans.

**Informed Consent Statement:** Informed consent was obtained from all subjects involved in the study.

**Data Availability Statement:** Our final map is available on the third-party repository of the Open Science Framework [79] at https://osf.io/7ztwu/, under 'sample maps'. Our map is named 'Giabbanelli & Macewan'. Other maps on obesity are provided for comparison and include the Foresight Obesity Map as well as the work of Verigin et al. [80].

**Acknowledgments:** We are indebted to Diane Finegood from Simon Fraser University. Her leadership in obesity research was instrumental in identifying experts in the area. We also appreciated her patience in providing feedback on the first semi-structured interview. Our report to the PHSA benefited from the feedback of Andrew Tugwell and Lydia Drasic from the BC Centre for Disease Control Operations and Chronic Disease Prevention with the PHSA. We also thank Ellen Lo from Healthy Families BC (PHSA), and Fred Popowich from Simon Fraser University, for logistical support.

**Conflicts of Interest:** The authors declare no conflicts of interest. The funders had no role in the design of the study, analyses, or interpretation of data. The funders approved the list of experts independently prepared by the research team prior to the interviews. The funders provided feedback on the writing for clarity and approved the publication of the results.

## Appendix A. Key Informants

The alphabetical list of key informants is provided in Table A1 below.

**Table A1.** List and fields of expertise for the key informants.

| Name | Position | Fields of Expertise |
|------|----------|---------------------|
| Geoff Ball, Ph.D. | Professor and associate chair of research, Department of Pediatrics, Faculty of Medicine & Dentistry, University of Alberta, Canada | Optimize obesity management and prevention for children and families, including via clinical trials or qualitative research |
| Katherine Cianflone, Ph.D. | Professor Emeritus, former Canada Research Chair on Adipose Tissue (Tier 1), Universite Laval, Canada | Adipose tissue metabolism, factors controlling fat, molecular basis of obesity |
| Jean-Pierre Chanoine, Ph.D., MD | Clinical Professor, Pediatric Endocrinologist, Department of Pediatrics, BC Children's Hospital, Canada. Secretary General of Global Pediatric Endocrinology and Diabetes (GPED) | Pediatric endocrinology, capacity building, access to medicine |
| Jean-Philippe Chaput, Ph.D. | Professor, Department of Pediatrics, University of Ottawa. Research Scientist, Healthy Active Living and Obesity Research Group CHEO Research Institute | Prevention and Treatment of Obesity in Children, Sleep health, Screen time, Physical activity |
| Jean-Pierre Després, Ph.D. | Professor, Department of Kinesiology, Faculty of Medicine, Universite Laval, Canada Scientific. Director, International Chair on Cardiometabolic Risk, Universite Laval. Innovation and Science Director, Alliance Santé Québec | Adipose tissue distribution, visceral obesity, type 2 diabetes, lipids, lipoproteins, cardiovascular disease, and their prevention through physical activity and healthy living |
| Jim Frankish, Ph.D. | Clinical Psychologist & Endowed Professor, School of Population and Public Health, UBC (passed away) | Nutrition education, health literacy, community capacity, healthy communities, and health promotion in primary care |
| Danijela Gasevic, MD, Ph.D. | Associate Professor and Head of Professional Education, School of Public Health and Preventive Medicine, Monash University, Australia | Chronic disease prevention, particularly regarding the effect of physical inactivity and sedentary behavior on health |
| Carolyn Gotay, Ph.D. | Professor Emeritus, founding Canadian Cancer Society Chair in Cancer Primary, School of Population and Public Health, UBC, Canada | Interventions to reduce modifiable cancer risk factors, quality of life in cancer patients and survivors |
| Michael Hayes, Ph.D. | Professor Emeritus and former Director, School of Public Health and Social Policy, University of Victoria, Canada | Health inequities, disability, public policy, obesity, health literacy, population health promotion |
| Terry Huang, Ph.D. | Distinguished Professor and Chair, Department of Health Policy and Management, City University of New York, USA | Chronic disease prevention, design and health, built environment, public-private partnerships, cross-cultural health |
| David Lau, Ph.D., MD | Professor Emeritus, Department of Biochemistry & Molecular Biology, University of Calgary, Canada. Former Chair of Diabetes & Endocrine Research Group and Director of the Julia McFarlane Diabetes Research Centre | Fat cell biology in health and obesity, development of insulin resistance in obesity and diabetes, and cellular mechanisms of diabetic vascular complications |
| Scott Lear, Ph.D. | Professor, Pfizer/Heart & Stroke Foundation Chair in Cardiovascular Prevention Research, Faculty of Health Sciences, Simon Fraser University, Canada | Cardiovascular disease prevention, population health, ethnic disparities |
| Gary Lewis, Ph.D., MD | Professor, Department of Medicine and Department of Physiology, University of Toronto. Director, Division of Endocrinology and Metabolism, University of Toronto | Whole body, integrative, physiological studies in humans |
| Pablo Monsivais, Ph.D. | Associate Professor, Department of Nutrition and Exercise Physiology, Elson S. Floyd College of Medicine, Washington State University, USA | Public Health, Epidemiology, Social Inequalities, Food and Nutrition |
| Kim Raine, Ph.D. | Distinguished Professor, School of Public Health, University of Alberta, Canada | How social conditions and people's behaviors (particularly food and eating behaviors) interact to transmit obesity and chronic diseases through social means |

**Table A1.** *Cont.*

| Name | Position | Fields of Expertise |
| --- | --- | --- |
| Arya Sharma, Ph.D., MD | Professor of medicine, chair in obesity research and management, University of Alberta. Founder and Scientific Director, Canadian Obesity Network | Evidence-based prevention and management of obesity and related cardiovascular disorders |
| John Spence, Ph.D. | Professor, Faculty of Kinesiology, Sport, & Recreation, University of Alberta, Canada | Benefits and determinants of physical activity and how physical inactivity and sedentary behavior are related to health |
| Tom Warshawski, MD | Associate Clinical Professor of Pediatrics, UBC. Chair of the Childhood Obesity Foundation. Former head of pediatrics, Kelowna General Hospital, Canada | Promoting Healthy Active Living in children and youth |
| James Woodcock, Ph.D. | Professor of Transport and Health Modelling, University of Cambridge, UK | Health impacts of changes to how we travel and how such changes might occur |

## Appendix B. Components of the Map

*Appendix B.1. Physical Factors*

There are multiple linkages between obesity and diseases that are commonly comorbid (occur at the same time) with obesity. It should be noted that diverse biological pathways link obesity to these comorbidities, even within categories (e.g., cardiovascular disease, respiratory disease, cancer). For example, it may be more useful to think of cancer as a variety of diseases that obesity impacts through distinct mechanisms, rather than a single disease. Furthermore, multiple pathways may even exist for one condition: for example, asthma-like symptoms can be caused by a reduction in airway size due to thoracic and abdominal adiposity, or they may come through inflammation due to the physiological effects of obesity. Consequently, particular attention is paid to identifying the multiple pathways at stake. We first summarize the impact of obesity on comorbidities, and then examine the impact of comorbidities on obesity (e.g., detrimental impact on physical activity). Finally, we address other physical factors that impact or are impacted by obesity (e.g., sleep duration) but are not considered as comorbidities.

*Appendix B.2. Comorbidities of Obesity*

Obesity causes many changes in the body, such as dyslipidemia, excess adipose tissue, and mechanical stress, which together can decrease physical well-being. Obesity results from a caloric imbalance, whereby the energy acquired from food is in excess of the energy spent on physical activities. Excessive food intake can lead to an abnormal amount of lipids in the blood, which is known as dyslipidemia. Because the storage and usage of lipids involve many parts of the body (e.g., fat tissue, liver, muscles), dyslipidemia contributes to several diseases such as cardiovascular diseases. We start by summarizing the map with respect to the physiological consequences of dyslipidemia. Another aspect of obesity is excess body fat, known as adipose tissue. Adipose tissue not only functions as a passive energy repository, it also actively releases hormones that can contribute to metabolic diseases, certain forms of cancer, and dyslipidemia. The consequences of excess, dysfunctional adipose tissue are addressed after dyslipidemia. Even in a metabolically healthy sub-population of obese patients (i.e., with normal glucose levels, lipid levels, insulin levels, and inflammatory factors) who do not experience dyslipidemia and adipose tissue dysfunction, obesity still has detrimental consequences. Indeed, the excess weight puts mechanical strain on lung function as well as on the joints. The mechanical consequences of excess body weight, such as respiratory and musculoskeletal diseases, are addressed last.

**Dyslipidemia** and its consequences are depicted in Figure A1, where the relationships are numbered for the descriptions that follow. Overeating, which is one of the drivers of obesity (DY1), can lead to dyslipidemia through excessive ingestion of fats and carbohydrates (DY2). These are carried in the blood through additional lipoproteins, which interact with cell surface receptors to generate factors that increase blood pressure (DY3), contributing to the development of hypertension in the long term (DY4). The balance of lipoproteins also changes, as dyslipidemia has more low-density lipoproteins (LDL) and less high-density lipoproteins (HDL). LDL molecules are commonly known as "bad cholesterol" as they transport fat into artery walls, creating fatty plaques (atherosclerosis, DY5) that can lead to blockage in a coronary artery, causing a heart attack (myocardial infarction, DY6), or an artery in the brain, causing a stroke (DY7). Finally, excess fat circulating in the blood has to be stored. Some individuals can increase their body fat sufficiently to store the excess, but others cannot cope and instead deposit fat on skeletal muscle or the liver (nonalcoholic fatty liver disease), which both lead to insulin resistance (DY8), causing type 2 diabetes (DY9). As noted by an interviewee, there is a wide variation in the ability to store excess fat:

> "I would die before I reached a BMI of 50 because I do not have the physiology, the genetics which would allow me to put on a lot of subcutaneous fat. Because to become massively obese you must have subcutaneous adipose tissue that has

tremendous ability to expand. And you're able to deal with chronic energy imbalance. Some of us just can't. We develop diabetes, we have cardiovascular events."

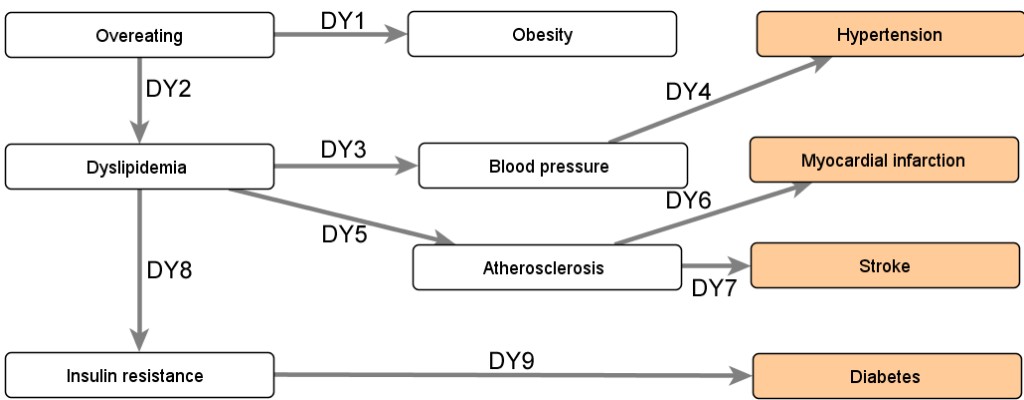

**Figure A1.** Consequences of dyslipidemia, with colored comorbidities on the right. All relationships depict an increase.

The consequences of **excess or dysfunctional adipose tissue** are summarized in Figure A2. Adipose tissue is primarily composed of adipocytes, which release adipokines. Obese individuals release more adipokines (AD1). Because many adipokines are atherogenic, this can lead to atherosclerosis (AD2), increasing the risk for heart attack and stroke, as highlighted previously (Figure A1). Furthermore, the adipose tissue of obese individuals may become dysfunctional, with immune cells going into the adipose tissue (AD3). One of the ways in which immune cells fight bacteria and viruses is by causing inflammation. However, obese individuals have a low level of chronic inflammation in the absence of harmful bacteria and viruses. Infiltration of macrophages, a type of immune cell, into the adipose tissue, causes harmful inflammation (AD4). Inflammatory proteins such as TNF-$\alpha$ can lead to insulin resistance (AD5), and inflammation can contribute (if the individual is genetically predisposed) to asthma (AD6) and autoinflammatory diseases (AD7). Similar to an autoimmune disease, an autoinflammatory disease results from the immune system attacking the body's tissue, which causes an increased state of inflammation (AD8). One such autoinflammatory disease is Crohn's disease, where there is ongoing inflammation of the intestinal tract. The inflammatory mediators released by the adipose tissue could play a role in linking obesity to cancer, however, due to limited physiological evidence in the literature, our map cannot currently detail the pathways from obesity to specific types of cancer (AD9). Dysfunctional adipose tissue also leads to dyslipidemia, possibly through the influence of macrophage infiltration on the lipoprotein profile or due to issues when trying to expand the visceral adipose tissue [100] (AD10). The adipose tissue may also overproduce angiotensinogen, which is part of the renin–angiotensin system that regulates blood pressure. Consequently, increased blood pressure is a possible outcome (AD11). Finally, it should be noted that obesity should not be thought of as simply increasing "everything" in the adipose tissue. For example, adiponectin is produced by the adipose tissue but it is downregulated for obese individuals. This downregulation still has an adverse health effect because it increases insulin resistance.

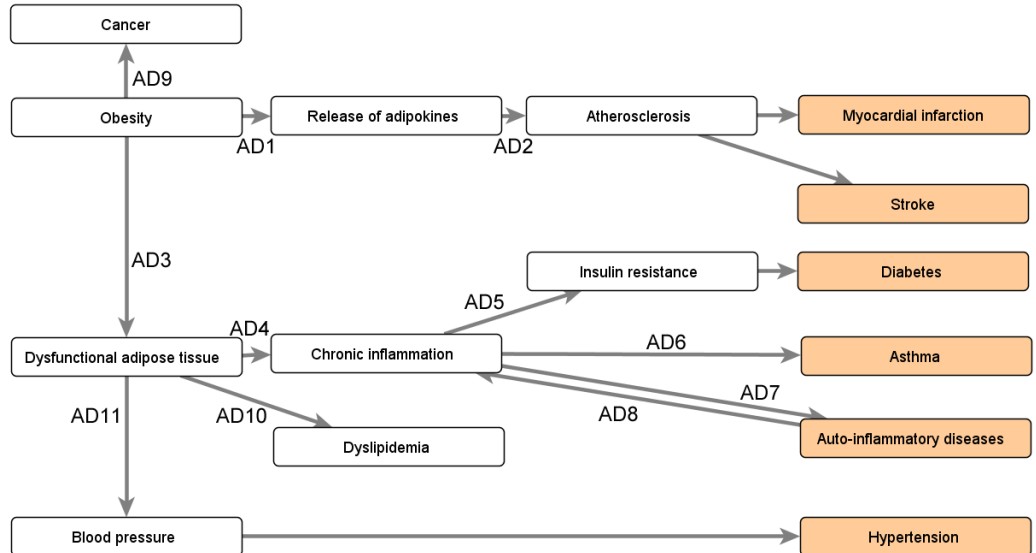

**Figure A2.** Consequences of en excess or malfunction of adipose tissue, with colored comorbidities on the right. All relationships depict an increase.

Obesity creates **mechanical** stress on the joints, particularly the knees, thereby contributing to osteoarthritis. Overweight and obesity are also known to contribute to chronic back pain [101,102]. Finally, obesity is involved in respiratory diseases such as asthma and obstructive sleep apnea through various pathways. Due to a lack of experts on respiratory disease in our panel, these pathways are resented in Figure A3, based on the seminal works of Sebastian [103] and Poulain et al. [104], complemented with recent perspectives [105,106].

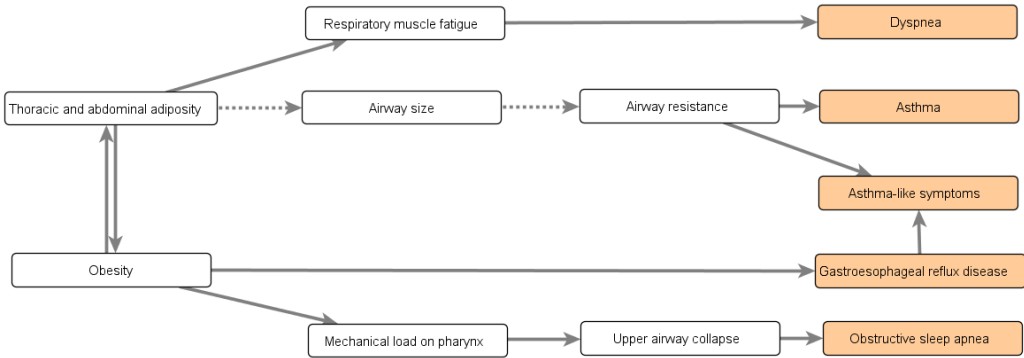

**Figure A3.** Pathways involved in linking obesity with respiratory disease based on Sebastian [103] and Poulain et al. [104]. Plain arrows indicate an increase whereas dotted arrows indicate a decrease.

*Appendix B.3. Impact of Obesity's Comorbidities*

Obesity's comorbidities create numerous biomedical **barriers to physical activity**. These barriers are depicted in Figure A4 and their relationships are numbered for the description that follows. First, many medications can limit one's ability to engage in physical activities, which is particularly relevant for older individuals who may be on multiple medications due to the presence of several comorbidities. Some of the medications limiting physical activities are:

- Beta-blockers (PH1), which reduce the heart rate, treating conditions such as hypertension (PH2), angina (PH3), or congestive heart failure (PH4);
- Medications prescribed for mental health problems (PH5), as they may affect concentration (PH6–7), coordination (PH8–9), or balance (PH10–11); and
- Medications causing tiredness (PH12–13), which are prescribed for a wide variety of issues ranging from depression to insomnia.

Second, pain can prevent engagement in physical activity (PH14). Pain can have widely different causes, such as angina (i.e., chest pain due to problems in the arteries) or musculoskeletal pain (PH15 and PH16 respectively). Third, cardiovascular diseases (CVDs) may affect physical activities but that relationship heavily depends on the extent to which the cardiovascular system is affected. For example, individuals with heart failure are much more limited in their daily activities than individuals with conditions such as hypertension. Interviewees emphasized that physical activity is nonetheless important for individuals with CVD because it will improve their condition in the long run. The main barrier created by CVD may not be physiological but rather psychological, interviewees suggested. Indeed, CVD may increase a person's fear (PH17–20), thereby making them afraid to engage in physical activities (PH21). Fourth, there may be issues in the locomotor system, such as joint issues typically associated with excess weight. However, such issues can be addressed by health professionals by recommending exercises that do not affect the joints (PH22). Finally, respiratory diseases for which obesity is an important risk factor (e.g., chronic obstructive pulmonary disease, asthma, obstructive sleep apnea, obesity hypoventilation syndrome) also limit physical activity (PH23). Nonetheless, physical activity is important for improving individuals' conditions, and adapted physical activities such as static exercises can still be performed.

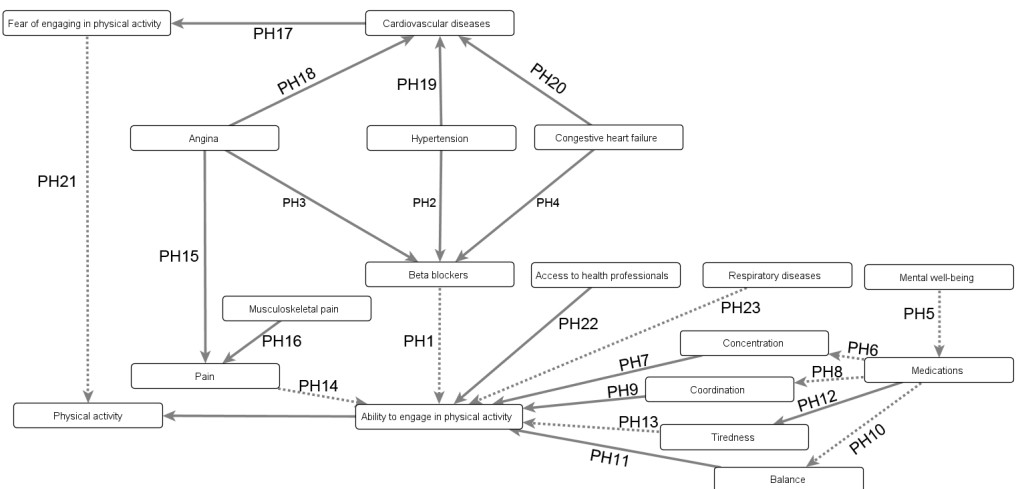

**Figure A4.** Consequences of obesity's comorbidities on fear and ability regarding physical activities.

Changes in **eating behavior** follow a pattern similar to the pathways through which impaired physical well-being creates barriers to physical activity. Relationships are summarized in Figure A5. For example, diabetes is commonly associated with obesity (EA1–2), and clinical trials have shown that medications used to treat diabetes (EA3) can impact weight in very different ways (Figure A4), creating a feedback loop. Several physiological pathways are involved (EA4), such as changes in leptin (which acts as a 'fullness' hormone to signal that the body has had enough food). It should be noted that the pathways involved are not only physiological. For example, sulfonylureas (SUs) are used to treat excessive blood sugar levels (hyperglycemia) but their primary mechanism acts irrespective of the blood sugar level; this can lead to hypoglycemia, which is a medical emergency. The UK Prospective Diabetes Study (UKPDS) showed that 27.8% of patients treated with a specific SU had hypoglycemic episodes, so it may be possible that patients taking SUs increase food intake or adopt 'defensive snacking' for fear of hypoglycemia [107] (EA5). Medications associated with the treatment of mental health conditions can also result in changes in appetite (EA6–9). A very short pathway can be involved, as is the case for anticonvulsants (used to treat mood disorders) and antipsychotics. Changes in appetite can also be triggered through a longer pathway, such as by affecting hormonal systems or sleep, which ultimately affects appetite. Changes in appetite that ultimately result in cravings (regardless of whether they were caused by medications) can be particularly

problematic as they can cause over-consumption of energy-dense foods, which pack many calories within each gram. Impaired mental well-being can also impact eating without the uptake of medications. Emotional eating is one example of the impact that poor mental well-being can have on food consumption. This relationship operates through a different system from the homeostatic system previously described in this section. Indeed, emotional eating works largely through the hedonic system and not the homeostatic system, which is the appetite system. Consequently, emotional eating involves a different set of signals, including endorphins and endocannabinoids (EA10). At a high level, there are two very harmful feedback loops. First, mental health conditions can increase weight by increasing appetite (as well as lowering physical activity) and increased weight can further decrease mental well-being. Second, an interviewee highlighted that if an individual who is facing economic barriers to healthy eating (EA11) needs expensive medication, they will have less disposable income for healthy food (EA12–14).

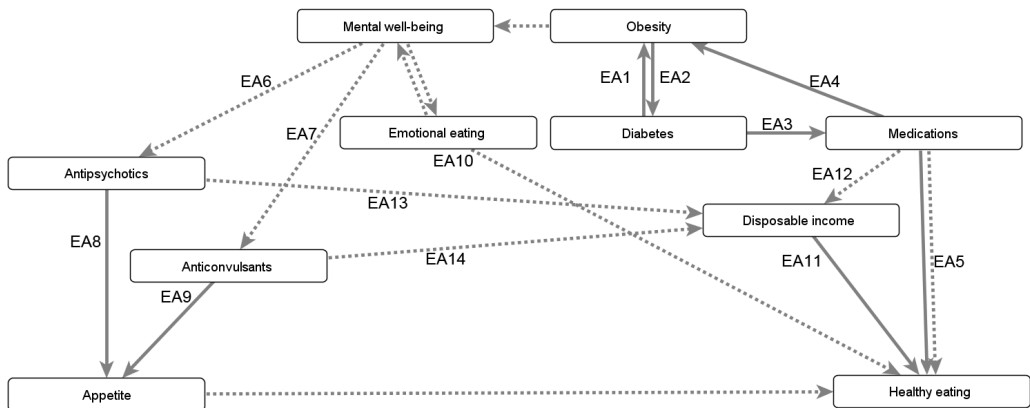

**Figure A5.** Consequences of obesity and mental well-being on healthy eating.

Appendix B.3.1. Mitigating the Negative Consequences of Obesity's Comorbidities

Interviewees stated that the **impact of physical activity** on weight is "much less than most people think", pointing to what they viewed as a "very wide-spread misconception that physical activity can somehow prevent obesity". At the same time, they acknowledged that exercise can have significant benefits in terms of physical and mental well-being, particularly when it is used to treat or prevent the detrimental consequences of obesity. Exercise is different from physical activity; it refers to the part of physical activity that is planned, structured, and repetitive. Figure A6 synthesizes several of the pathways that are involved. Eating was included for the sake of completeness as it can impact the action of physical activity, sometimes counter-balancing its benefits (e.g., physical activity can decrease blood pressure, but this effect can be countered by high sodium intake in the diet). It should be noted that the large number of factors related to the cardiovascular system in Figure A6 is a consequence of having numerous interviewees with expertise in this system. Physical activity also impacts other systems not depicted in this figure. For example, it relieves stress and lowers the associated cortisol levels that would otherwise have a negative impact on healthy eating. A nuanced understanding still needs to be achieved regarding other diseases. For example, the impact of physical activity on cancer (which is part of obesity's comorbidities) depends on the cancer site [108,109], and reviewing all types of cancer is beyond the scope of our map.

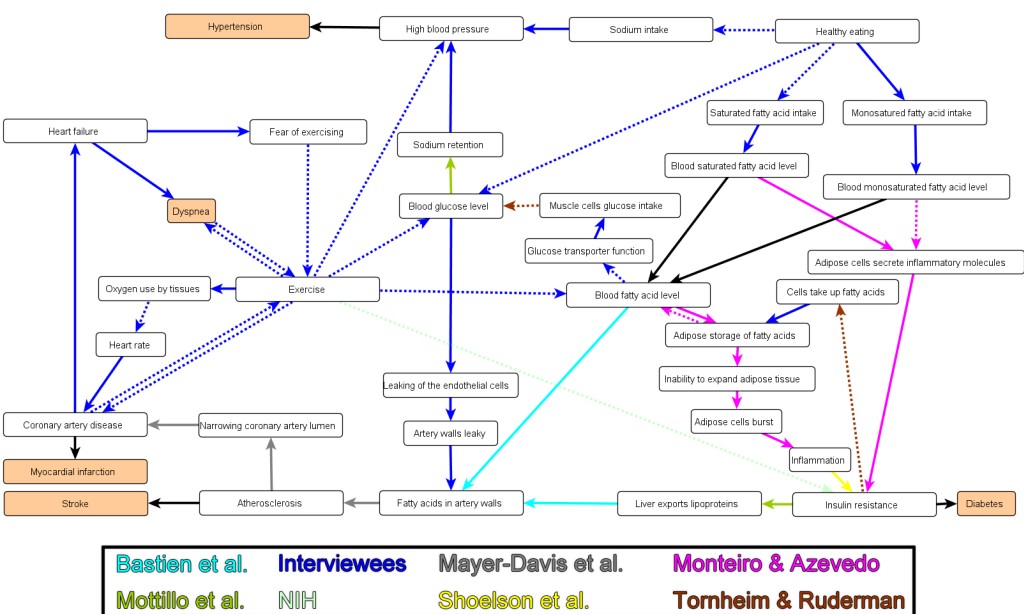

**Figure A6.** Pathways linking exercise and healthy eating to comorbidities. Factors and directed relationships were mostly based on the interviewees' synthesis of the evidence (54% of relationships). Other sources include the highly cited work of Monteiro and Azevedo [110] (14%), the NIH's 2022 updated definition of atherosclerosis [111] (12%), and a chapter by Tornheim and Ruderman [112] (10%). The remaining components come from the highly cited overview of Bastien et al. [113], a classic study by Mayer-Davis et al. [114], the authoritative review of Mottillo et al. [115], and the well-known work of Shoelson et al. [116]; previous subsections completed this subsystem (black arrows). Relationships causing an increase are depicted by plain arrows, while those causing a decrease are depicted by dashed arrows.

Appendix B.3.2. Sleep Duration

Short sleep duration is an important contributor to obesity as well as to several comorbidities of obesity. These relationships are summarized in Figure A7. Short sleep duration prevents the recovery of the hormonal profile that regulates appetite (SL1). Specifically, short-duration sleepers have lower plasma leptin levels and higher ghrelin levels than predicted by their body mass, which translates into lower satiety and higher hunger, respectively. Consequently, short sleepers eat more (SL2). This is reinforced by the fact that, since individuals are awake for more hours during the day, they also have more time to eat (SL3). Short-duration sleepers have added negative consequences when they also engage in late-time TV viewing: there is a direct effect whereby adults' excess caloric intake at the end of the day is associated with screen time (SL4), and an indirect effect, given that the light of the TV suppresses melatonin that regulates circadian rhythms (SL5). As in many health issues surveyed in this section, short sleep duration not only increases obesity but can also be increased by it (SL6). For example, obesity is a driver of obstructive sleep apnea, which negatively impacts sleep. It should be noted that a U-shaped relationship was proposed in the mid-2000s between sleep and weight gain, suggesting that both short and long sleep durations could favor weight gain. However, this relationship was suggested based on self-reported sleep. Recent studies using objective measures of sleep (e.g., wearing a watch that measures sleep time) did not find this U-shaped relationship. Numerous other factors affect sleep (SL7), such as the intake of stimulants (e.g., coffee, energy drinks) but a complete list is beyond the scope of this report.

Short sleep duration has direct negative consequences on the body, in addition to the aforementioned hormonal changes in ghrelin and leptin. A lack of sleep is a stress factor, which is witnessed by the increase in cortisol in the same way as when experiencing mental stress (SL8). A lack of sleep increases the sympathetic nervous system activity, which impacts many organs. For example, it raises blood pressure (SL9) and accelerates

heart rate (SL10), contributing to hypertension and coronary heart diseases, respectively. Epidemiological evidence also shows that short sleep duration is associated with elevated pro-inflammatory cytokines (SL11) [117], and chronic inflammation can lead to type 2 diabetes [118]. Our map only showed a relationship between sleep duration and energy intake (i.e., eating) but it did not include a relationship with energy expenditure. The impact of short sleep duration on energy expenditure is limited and subjected to a large variation between individuals. While short sleep duration increases tiredness, it does not limit physical activity in all individuals. Moreover, the body consumes more energy when it is awake (i.e., higher basal metabolic rate) than when sleeping. Interviewees emphasized that the link between a lack of sleep and weight gain was primarily due to an excess in food intake.

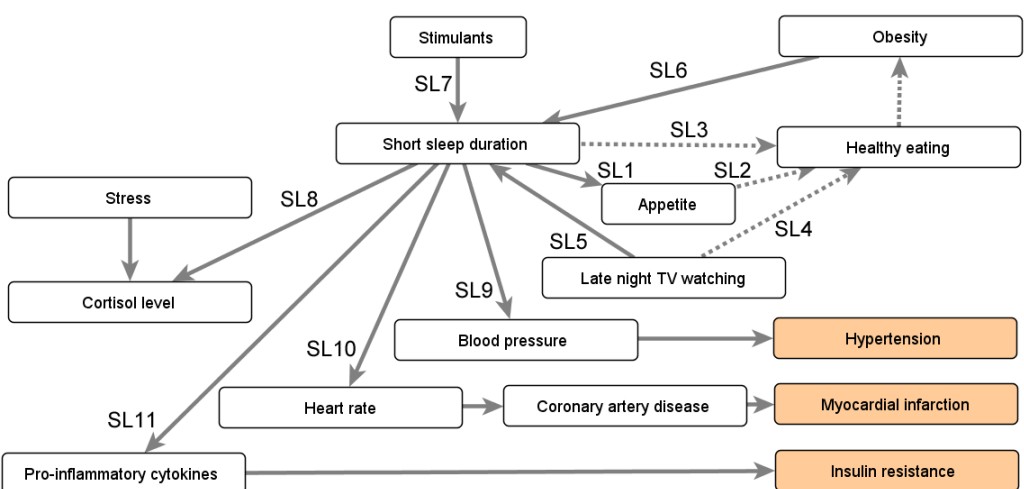

**Figure A7.** Relationships between short sleep duration, obesity, and comorbidities of obesity. Relationships causing an increase are depicted by plain arrows, while those causing a decrease are depicted by dashed arrows.

*Appendix B.4. Environmental Factors*

A variety of resources are necessary to support individuals in participating meaningfully in the society in which they find themselves, such that they can achieve a state of physical and mental well-being. Resources include tangibles, such as physical access to healthy food retailers, and intangibles, such as being part of a community that is accepting of different body shapes. Two types of resources are of particular interest in the map: the *built environment* and the *social environment*. The *built environment* involves the set of human-made surroundings in which activities of daily life take place. Measures of the built environment can be objective, such as the presence of sidewalks or food stores, or perceived, such as fear of crimes. In this work, we define the *social environment* in a broad sense: it is the set of social norms and human interactions. For example, an interviewee expressed that individuals who know their neighbors will have better mental well-being since they are less likely to perceive their neighborhood as dangerous. The built and social environments are tightly linked. In the words of an interviewee, "when we are talking about the built environment, I would never separate it from the social environment". Consequently, our analyses address both built and social environments without attempting to artificially separate them.

Appendix B.4.1. Influence of the Built and Social Environments on Eating Behaviours

The impact of the built environment on eating is depicted in Figure A8. The relationships are numbered for the description that follows, starting from the influence of a supportive food environment on healthy eating (BE1). A supportive food environment makes it easier for people to choose healthy dietary options. Three key factors influence the supportive food environment: food cost, marketing, and availability. The first factor

is the ratio of the cost of unhealthy food to healthy food (BE2). For example, "when it is much cheaper to buy pop and chips than to buy milk and apples, then that is definitely not making the healthy choice the easy choice". The second factor is the ratio of marketing for unhealthy food compared to healthy food (BE3). An interviewee highlighted that "because marketing is usually done to make things seems cool and fun and all of that type of thing, it's not usually the healthy foods that are being marketed". The third factor is the ratio of the availability of unhealthy food to healthy food (BE4). For example, this compares the number of fast food outlets or convenience stores with the number of stores selling fresh fruit and vegetables within walking distance.

All three factors are in turn impacted by other contributors. Only public support and demand for healthy products were mentioned as having the potential to improve all three factors, by making healthy food products more available (BE5), affordable (BE6), and potentially shifting the balance of marketing (BE7). This potential should nonetheless be taken with a note of caution, as an interviewee highlighted that defining a healthy product can be difficult, and that the potential of directing taxes at the government level to shift what is affordable has faced several issues. Other aspects of the environment mostly impact a single factor. Consumer demand for advertisement regulations could contribute to reducing the amount of advertising for unhealthy food products, thereby impacting the ratio of marketing for unhealthy food compared to healthy food (BE8). Evidence in support of this suggestion includes the Quebec ban on advertising unhealthy food to children. This ban reduced the probability of fast food purchase incidence by an estimated 13% weekly29. The existence of restrictive covenants negatively impacts the availability of healthy food products (BE9), as supermarkets that relocate have to be replaced by stores that do not sell food, thereby creating areas without adequate healthy food options. Finally, economic incentives to produce unhealthy food products contribute to the low cost of unhealthy food compared to healthy food (BE10).

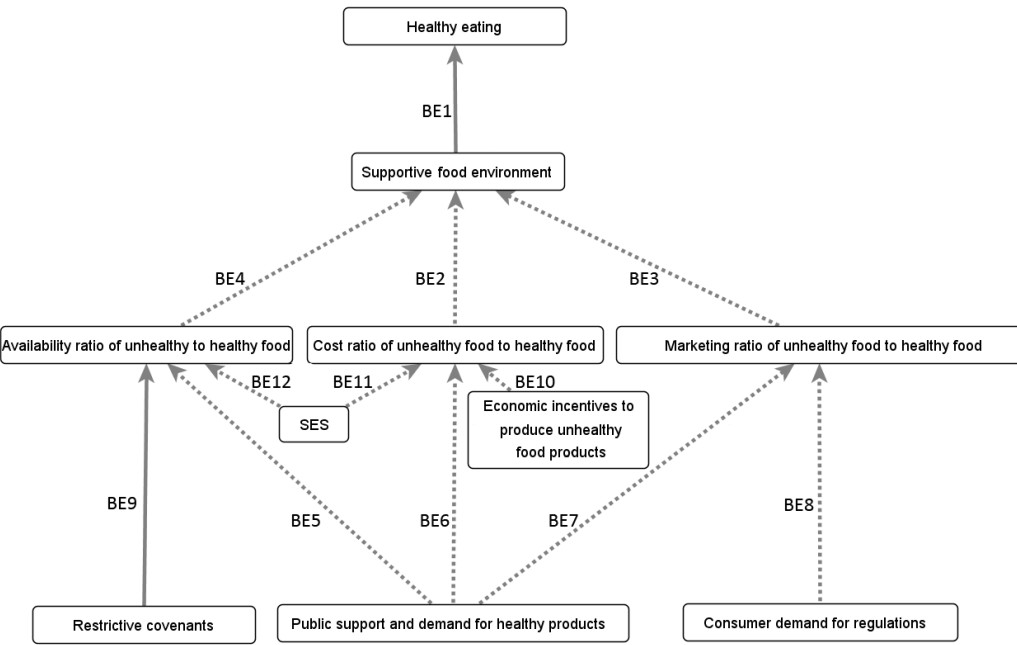

**Figure A8.** Relationships between the objective food environment and healthy eating. Relationships causing an increase are depicted by plain arrows, while those causing a decrease are depicted by dashed arrows.

Socioeconomic status can impact the individual's ability to access food in their environment, thus influencing the quality of their diet. Healthier diets are relatively more expensive, and this is more of a barrier to healthy eating for individuals of lower socioeconomic status (BE11). Interviewees tended to agree that individuals with a higher SES

generally inhabit neighborhoods whose characteristics promote healthier choices and offer better access to healthy foods (BE12). One interviewee also discussed research in the US that found that lower-income individuals tended to shop further from home in order to find food within their price range, often passing several supermarkets on their way. In summary, the availability of healthy food "is likely to involve not just the physical proximity, but what you can afford when you step in the door", which further motivates linking SES to the presence of healthy food retailers over unhealthy ones (BE12). Furthermore, socioeconomic status can impact an individual's perception of the objective built environment. For example, a low SES individual may believe healthy grocery stores are too expensive and, therefore, stop perceiving the healthy options as part of their food environment.

Appendix B.4.2. Influence of the Built and Social Environments on Physical Well-Being

Social norms constitute an important component of the social environment. The social norm of weight stigma impacts how overweight individuals perceive, and ultimately interact with, their environment. Individuals who experience weight stigma or bullying may not perceive their environment as inviting or friendly; instead, stigma creates fear of interacting with people in the social environment, which an interviewee summarized as "our environment becomes our enemy". This unfriendly environment can prevent overweight individuals from engaging in physical activity. Indeed, this social stigma and its negative impact on mental well-being may create a larger barrier to physical activity and, thus, to physical well-being, than weight status itself. In the words of an interviewee, "a lot of people dealing with weight issues have had traumatic experiences when it comes to physical activity" due to weight bullying.

Social norms such as weight stigma function at a society-wide level. At a local level, the social environment is also shaped by communities, which play an important role in determining how individuals interact with and perceive the built environment. For example, engaging in physical activity with members of one's community can help create a routine around these activities such that physical activities become sustainable in the long term. Because of the important role communities play in determining interactions with the built environment, using the community's input when planning changes to the built environment can be very valuable, to ensure modifications are made in a manner relevant to the identity and culture of the community. This will ultimately affect whether individuals use the infrastructure that is made available. In the words of an interviewee:

> "As soon as we build it and if we think it's really important, we should promote it. [...] But before it is built, and that's the biggest mistake now, things are often built without consulting with the community. [...] What are the needs of the community? Because gyms may be built in communities where people are of low-income and cannot afford to go to the gym, so what is the use of it? [...] Maybe they would prefer [...] children's playground or playing fields where kids could play."

The idea that social norms influence interactions with the environment is met by a symmetrical relationship: the environment can lead to changes in social norms. This pathway works through several layers, which we do not discuss in this report as they have been clarified elsewhere (e.g., by Macmillan and Woodcock in the case of cycling [119]), and our map provides a high-level understanding of obesity and well-being.

When examining the specific components of the built environment that contribute to physical activity, it is useful to categorize physical activity into several groups depending on the purpose. In utilitarian physical activity, the goal is to move from one place to another (e.g., walking to reach a store); this is also known as active transportation. In recreational physical activity, individuals engage in activities for enjoyment or pleasure; this is also known as leisure-time physical activity. While there are other forms of physical activity (e.g., work-related physical activity, household-related physical activity), these are beyond the scope of the current report. The relationships surveyed for both utilitarian (AC1) and recreational (AC2) physical activity are summarized in Figure A9.

Interviewees highlighted three barriers to recreational physical activity: a lack of time (AC3), a lack of infrastructure (e.g., the presence of green fields and sports fields, walking trails, and bike trails) (AC4-7), and a lack of positive sports experiences (AC8). Regarding utilitarian activities, micro-environmental factors (e.g., aesthetics, sidewalk prevalence and maintenance, and street crossing features) (AC9-11) as well as accessibility to shops (AC12) were two commonly mentioned features. Perceived safety was considered to be relevant for both activity types (AC13), as fear can prevent individuals from engaging in physical activity in their neighborhood regardless of purpose. As discussed above, weight stigma can reduce the positive experience of sports (AC14), for example, one may not feel at ease in a swimsuit depending on one's body shape.

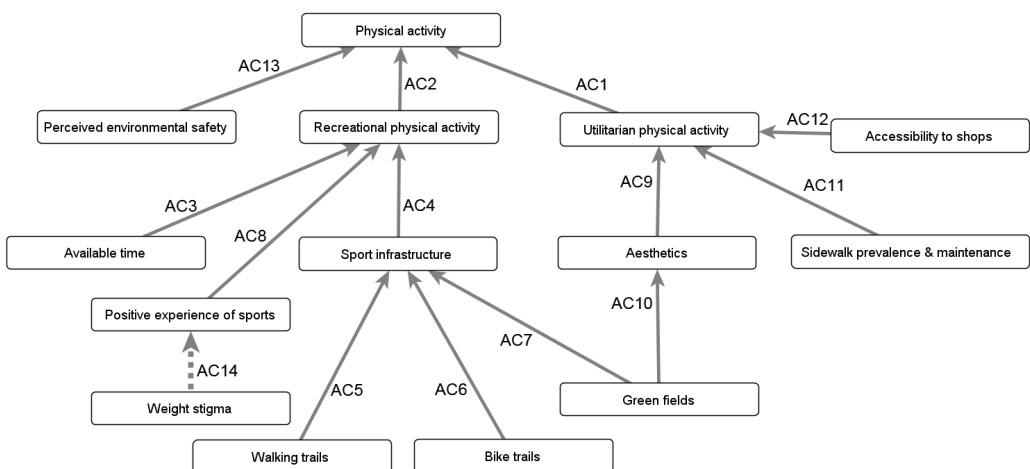

**Figure A9.** Barriers to physical activity. Relationships causing an increase are depicted by plain arrows, while those causing a decrease are depicted by dashed arrows.

Studying the economics (e.g., costs associated with recreational activities) or specific types of physical activity is beyond the scope of the present report and comprehensive models have been developed elsewhere. For example, a series of diagrams was recently released regarding key factors and interrelationships in physical activity [119,120]. The case of cycling provides a good illustration of one of the core loops in physical activity: the loop of perceived danger. The more perceived danger, the more safety equipment is used by people. The more equipment is used, the more individuals perceive the environment as dangerous.

*Appendix B.5. Factors Not Included*

Several factors and relationships were not included in the map due to concerns raised during the interviews, a lack of evidence, or because they did not focus on obesity and well-being in the general Canadian population. For full disclosure, we briefly cover such constructs, starting with 'knowing the healthy choice' to physiological factors and pollution.

We assessed whether **healthy food choices** were partly driven by knowledge regarding healthy eating. This possibility was endorsed by none of the interviewees, who instead provided several reasons to reject it. First, interviewees suggested that "people have a good sense of what is healthy and what is not healthy". That is, individuals are generally not limited by a lack of knowledge regarding healthy eating. Second, interviewees felt that there are more important barriers than knowledge when it comes to food choices, because "healthy choices are not the easy choices: they tend to be the more expensive choices and they tend to be the less satisfying choices". Consequently, a key issue may be a lack of resources to act on one's knowledge of healthy eating; these constructs are covered in the map. Finally, interviewees mentioned that one's knowledge can still be trumped by food marketing, such that point-of-purchase decisions are also heavily influenced by how foods are promoted and made attractive, which are also covered by our map. We further examined

whether knowledge of the healthy choice was impacted by formal education. Should that relationship hold, it could still have been valuable to include knowledge as a factor. However, all interviewees but one dismissed this possible relationship. The interviewee who did not dismiss the relationship outright stated that formal education might lead to increased knowledge of healthy options; however, they did not see this increase in knowledge as having an important impact on dietary quality. The interviewees who dismissed the relationship provided counter-examples by which individuals with a high level of formal education still engage in unhealthy eating, while individuals with a low level of formal education actively seek healthier options.

Our map includes many **physiological** factors, but some were not included: sarcopenia, mitochondrial dysfunction, pregnancy, early fetal development, and the intestinal flora. We examined the role of *sarcopenia*, which is a loss of muscle mass. One interviewee detailed that sarcopenia can happen for very different reasons: while sarcopenia is more likely to be seen in older sedentary individuals than younger ones, it can also be fueled by a chronic inflammatory disease, and has been observed in post-cancer patients. Furthermore, these causes need to be investigated in detail: while obesity can cause inflammation, simply being obese does not lead to sarcopenia in most cases because obesity requires more muscle to support the increased body mass. Consequently, we concluded that additional research on the role of sarcopenia was needed. Within our assessment of the consequences of obesity in terms of comorbidities, one interviewee discussed the role of *mitochondrial dysfunction*. Mitochondria generate energy for cells through the oxidation of glucose and fatty acids. One of the markers of insulin resistance and type 2 diabetes is the inability to switch between glucose or fatty acids as the primary source for the generation of energy, which is known as metabolic inflexibility. Consequently, many studies have investigated the role that mitochondrial dysfunction could play in insulin resistance and type 2 diabetes. Overall, there is strong evidence for a correlation between mitochondrial dysfunction and the development of metabolic diseases but revealing the causal link remains a challenge (see [121] for an updated review). Several mechanisms have been proposed. For example, a high level of fatty acids could increase their rate of oxidation; since fatty acid and glucose oxidation are reciprocally regulated, that could reduce glucose oxidation and prompt the muscles to stop 'listening' to the insulin signal to take up glucose for energy, thereby making the muscles insulin resistant [122]. However, this theory of impaired insulin signaling through the over-accumulation of fats is debated, and further research is needed before representing such theories in the map.

We also investigated *pregnancy and early fetal development*. These are often discussed in relation to the prevention of early childhood obesity and epigenetics. During fetal development, the organism can adapt to environmental factors; its evolutionary-determined gene expression will undergo long-term changes by altering the packaging of DNA (epigenetic modification). For example, fetal exposure to excess blood lipids could alter the genes involved in lipid sensing and metabolism. In addition to the conditions that offspring may develop, several interviewees highlighted that pregnancy also has important consequences on the mother through postpartum weight retention. While the average pregnancy-associated weight gain is small at the population level [123,124] (under 2 kg), there are still between 14% and 25% of women who retain at least 5 kg a year after giving birth [125]. Consequently, pregnancy is a risk factor for being overweight [126], particularly for the deposition of fat in the abdominal region [125]. This relationship runs both ways, as pre-pregnancy weight also has a strong influence on postpartum weight retention [123]. The mechanisms of postpartum weight retention are highly complex: they involve (i) the characteristics of the mother before pregnancy (e.g., weight), (ii) the development of the pregnancy (e.g., weight gain early in pregnancy leads to a higher risk of weight retention [123]), and (iii) events unfolding after giving birth (e.g., postpartum depression). Numerous physiological pathways may also be involved, given the hormone changes after pregnancy. For example, a lower production of estrogen and progesterone can lead to lower serotonin levels or an excess secretion of cortisol [125], both having consequences on

food intake and the deposition of fat within the body. Postpartum weight retention is not included in the current map because the map is neither age- nor gender-specific. However, the importance and complexity of postpartum weight retention would make it a key factor if the map were to be tailored to young women.

One interviewee emphasized the importance of new developments in research on *intestinal flora*, highlighting that studies are still needed. The interviewee suggested that "it may be as important as the built environment because of its modifiable aspect throughout life" since individuals can be exposed to different bacteria (e.g., yogurt and a low-calorie chocolate bar may have the same calories but different bacteria) while being possibly predisposed to one type of flora. This should also be put in relation to early development. For example, a cesarean section prevents the newborn from being in contact with the flora of the mother, which will affect how the baby's gut is colonized and can metabolize food. In addition, new studies are suggesting possible links to mental well-being, independent of obesity, through the types of metabolites generated, absorbed, and potentially affecting the brain.

Finally, **pollutants** could be involved in several interactions that are not represented in the map. There is a relationship between pollutants and physical activity since a change in commuting patterns (e.g., from driving to walking or cycling) could lower car emissions, thus reducing the air pollution that contributes to respiratory diseases (several of which are part of obesity's comorbidities). However, this effect would be counter-balanced because individuals who engage in active modes of travel breathe more pollutants than car drivers due to a higher ventilation rate required during physical activity. Furthermore, interviewees concluded that the adoption of active travel would have to be massive before it makes a noteworthy difference in air quality, and air quality itself was not one of the dominant factors influencing the uptake of active modes of transportation within the Canadian setting. Consequently, we did not expand on this relationship in our proposed map. Furthermore, interviewees suggested that pollutants could contribute to some of the comorbidities of obesity. Several compounds used as pesticides (e.g., organochlorines) have a high affinity for fat. Given that obese individuals are characterized by an excess of body fat, they can, thus, have more pesticides in their bodies. This is not necessarily a problem when, in the words of one interviewee, "pollutants are sleeping in the fat tissue". However, pollutants are released in the plasma during weight loss, and this increased plasma concentration of pollutants could affect the body. Exploring the consequences is still the subject of ongoing research.

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
