# Peer review of "Leveraging Artificial Intelligence and Participatory Modeling to Support Paradigm Shifts in Public Health: An Application to Obesity and Evidence-Based Policymaking"

_information, doi:10.3390/info15020115_

Round 1
Reviewer 1 Report
Comments and Suggestions for Authors
Dear Editor
Attached to this email you could find my comments for the following paper:
" Leveraging Artificial Intelligence and Participatory Modeling to Support Paradigm Shifts in Public Health: An Application to Obesity and Evidence-Based Policymaking”.
Please do not hesitate to contact me for any further questions.
Kind regards

-
Author Response
Thank you for your feedback. Please find attached a letter in which we have responded to each point.

Reviewer 2 Report
Comments and Suggestions for Authors
I read the article with considerable interest as it fits this journal's purpose. The subject, in my honest view, is intriguing enough to draw readers in. The manuscript is properly organized, and while most of the sentences are clear, there are no sporadic odd sentence constructions. The study is easier to read overall thanks to its high quality. Although English is generally utilized appropriately in terms of grammar and syntax. The important contribution closes the gaps that could lead to significant, difficult, and exciting new research avenues.
I recommend this article be published
Author Response
We appreciate the kind and supportive words of the reviewer. We are glad to hear that the reviewer shares our vision for the exciting research venues that could stem from this work.